

# Submesoscale $CO_2$ variability across an upwelling front off Peru

Eike E. Köhn[1], Sören Thomsen[1], Damian L. Arévalo-Martínez[1], and Torsten Kanzow[2]

[1]GEOMAR Helmholtz Centre for Ocean Research Kiel, Kiel, Germany
[2]Alfred-Wegener-Institute Helmholtz Centre for Polar and Marine Research, Bremerhaven, Germany

*Correspondence to:* Eike E. Köhn (ekoehn@geomar.de)

**Abstract.** While being a major source for atmospheric $CO_2$, the Peruvian upwelling region exhibits strong variability in surface $f\mathrm{CO_2}$ on short spatial and temporal scales. Understanding the physical processes driving the strong variability is of fundamental importance for constraining the effect of marine emissions from upwelling regions on the global $CO_2$ budget. In this study, a frontal decay on length scales of $\mathcal{O}(10\ \mathrm{km})$ was observed off the Peruvian coast following a pronounced
decrease in downfrontal wind speed with a time lag of 9 hours. Simultaneously, the sea-to-air flux of $CO_2$ on the inshore (cold) side of the front dropped from up to 80 to 10 $\mathrm{mmol\ m^{-2}\ day^{-1}}$, while the offshore (warm) side of the front was constantly outgassing at a rate of 10-20 $\mathrm{mmol\ m^{-2}\ day^{-1}}$. Based on repeated ship transects the decay of the front was observed to occur in two phases. The first phase was characterized by a development of coherent surface temperature anomalies which gained in amplitude over 6-9 hours. The second phase was characterized by a disappearance of the surface temperature front within 6
hours. Submesoscale mixed layer instabilities were present but seem too slow to completely remove the temperature gradient in this short time period. Dynamics such as a pressure driven gravity current appear to be a likely mechanism behind the evolution of the front.

## 1 Introduction

Although the ocean generally acts as a net sink for atmospheric $CO_2$ (Takahashi et al., 2009), most low latitude eastern bound-
ary upwelling systems (EBUS) are natural sources of $CO_2$ to the atmosphere (Chavez et al., 2007; Capone and Hutchins, 2013; Gruber, 2015). The distribution of $CO_2$ in such areas is complex and results from the interaction betweeen cooling/warming at the surface, upwelling and mixing, biological activity, and riverine carbon and nutrient input (Laruelle et al., 2014; Gruber, 2015). As one of the four major EBUS, the Peruvian upwelling region is an important area for the exchange of climate relevant gases (e.g. $CO_2$, $N_2O$) between the ocean and the atmosphere (Friederich et al., 2008; Arévalo-Martínez et al., 2015). Hence,
quantification of sea-to-air fluxes of $CO_2$ as well as their spatial and temporal variability is essential for constraining the global budget of $CO_2$ in a changing climate.

The surface $CO_2$ in the Peruvian upwelling region is highly variable on short time $\mathcal{O}(\text{hours-days})$ and space $\mathcal{O}(\text{km})$ scales (Friederich et al., 2008), due to both biological and physical processes. The sharp lateral temperature gradient, separating the newly upwelled, cold water from warm surface waters further offshore, allows for pronounced frontal processes, which
manifest themselves in eddies and filaments (Penven et al., 2005; Chaigneau et al., 2008; McWilliams et al., 2009; Colas et al., 2012; Thomsen et al., 2016a). These submesoscale features, which go along with Rossby numbers $Ro = \mathcal{O}(1)$, can develop



strong vertical velocities in the upper ocean layer. Thus, surface fronts may enable the exchange of large amounts of heat and gas between the atmosphere and the subsurface ocean (Ferrari, 2011). However, the small spatial and temporal scales involved make submesoscale frontal processes difficult to observe. At the same time model studies have put forward their importance for physical-biogeochemical coupling, e.g. by alterating the vertical transports of nutrients and organic carbon (Lapeyre and

Klein, 2006; Lévy et al., 2012).

The link between the surface $CO_2$ distribution and the (sub-) mesoscale flow field was studied in the open ocean of the Northeast Atlantic by Merlivat et al. (2009), using both underway and lagrangian surface drifter measurements. During the measurement period, the study area was characterized by weak eddy kinetic energy. Still, submesoscale variability with large amplitude variations of surface $CO_2$ concentrations on horizontal length scales of $\mathcal{O}(10\ \mathrm{km})$ was observed. This variability

was successfully reproduced by the modelling study of Resplandy et al. (2009) but it does not seem to have a major effect on the model's overall $CO_2$ budget. The influence of submesoscale variability on the overall $CO_2$ budget might be stronger in the case of EBUS due to the ubiquitous existence of sharp fronts and filaments, i.e. in the case of a highly energetic (sub-) mesoscale flow field (McWilliams et al., 2009; Colas et al., 2012). However, so far no observations are available which describe the variability on the submesoscale off Peru.

In this study repeated measurements of $CO_2$ and physical properties across the upwelling front off Peru near 14° S are presented (see Fig. 1 for the large scale setting of the experiment). Throughout the two day experiment, we observed short-term fluctuations (timescales on $\mathcal{O}(\mathrm{hours})$) of the surface temperature and velocity field. Simultaneously, surface $CO_2$ sea-to-air fluxes showed pronounced changes, implicating the importance of these timescales for the ocean-atmosphere gas exchange. The goal of this paper are to: 1) document the high-frequency variability across the upwelling front and 2) discuss possible

physical driving mechanisms, improving our current understanding of the variability of surface $CO_2$ in the Peruvian upwelling region. The field work was conducted during the RV *Meteor* cruise M93 in February/March 2013, and was carried out within the framework of the Research Collaborative Centre SFB754 (www.sfb754.de).

This paper is structured as follows: In section 2 we present the used physical and biogeochemical datasets used for this study, as well as the methods employed for their analysis. Section 3 contains a description of both the experiment and the initial state

of the front. Observations from the subsequent frontal evolution are presented in section 4. Subsequently, the changes in the temperature front are analyzed in the context of various possible underlying dynamics, such as surface heating (Section 4.1), Ekman buoyancy fluxes (Section 4.2), submesoscale mixed layer instabilities (Section 4.3) or pressure driven gravity currents (Section 4.4). In section 5 these mechanisms are compared with respect to their associated buoyancy fluxes. Section 6 contains a discussion of the different mechanisms and their likelihood in driving the observed variability. The conclusions drawn from

this study follow in section 7.





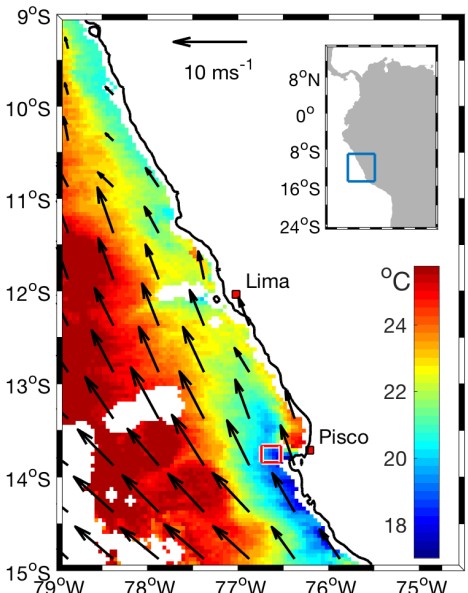

**Figure 1.** MODIS SST (color) and ASCAT (Advanced Scatterometer) wind field (arrows) off the coast of Peru on February 16, 2013. The small red box shows the study region off the coast of Pisco.

## 2   Data and methods

### 2.1   Hydrographic and meteorological measurements

The physical state of the upwelling front is mainly deduced from underway temperature, wind and velocity measurements. Underway temperature was measured by the thermosalinograph using an external SBE-38 digital thermometer at 3.5-4 m depth at the vessel's port-side bow. Temperature data was gathered at 0.1 Hz and filtered by a $2^{nd}$ order lowpass butterworth filter with a cutoff period of 250 seconds to remove the effect due to high frequent ship movements. Wind velocity and direction were measured at a height of 35.5 m above the sea surface at a temporal resolution of 60 second. The surface wind stress was estimated from these measurements following Smith (1988). Underway ocean current velocity measurements with a vertical resolution of 4 m were obtained from a vessel-mounted 75 kHz ADCP (vmADCP). The shallowest vmADCP based current velocities were obtained in a bin centered at 11 m below the sea surface. The data was averaged in 1-minute-ensembles and smoothed using a 3-minute running mean. A $16.56°$ counterclockwise rotation is applied to transform the measured currents into along-front and cross-front velocities (note the frontal orientation in Fig. 2). As a result positive cross-front velocities are directed towards the coast and positive along-front velocities indicate an equatorward flow. Underway measurements of salinity are unavailable as the thermosalinograph's conductivity sensor failed to produce consistent results. Likewise, temperature and vmADCP measurements taken during periods of highly variable vessel speed and heading proved to be unreliable due to the




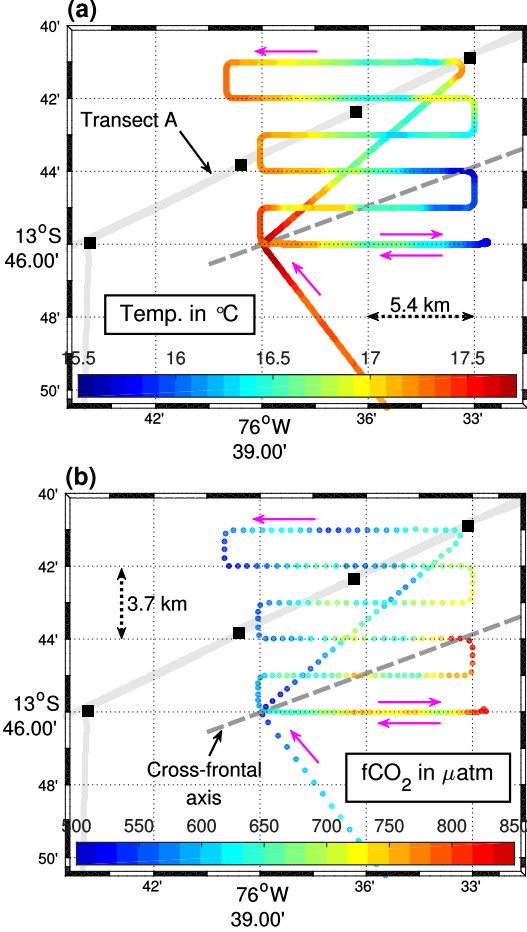

**Figure 2.** Sampling map of underway temperature (a) and $f$CO$_2$ (b) along the cruise track. The solid grey line shows the CTD section (transect A) conducted prior to zonal transects, with black squares indicating CTD stations. The dashed grey line depicts the cross-frontal axis, along which the front was crossed 17 times afterwards. These crossings include the high-resolution subsurface temperature transects B and C. Magenta arrows show the direction of the ship's track. The area presented in both panels is depicted as a red square in Fig. 1.

influence of the ship. Thus, current measurements immediately before and after CTD stations are neglected in the following data analysis. During the analyzed transects the vessel speed was held at approximately constant $4\,\mathrm{m\,s^{-1}}$ in order to achieve high quality ADCP data.

Hydrographic data *below* the surface were obtained from lowered conductivity, temperature and pressure (CTD) measure-
5 ments. These are arranged in transect "A" consisting of four shallow CTD profiles located $7\,\mathrm{km}$ apart (black dots in Fig. 2). The CTD was also equipped with a fluorometer and an oxygen sensor, allowing for chlorophyll-$a$ (Chl-$a$) and dissolved oxygen (O$_2$) measurements. A detailed description of the data processing, inluding the calibration of oxygen can be found in Thomsen et al. (2016b). As the Chl-$a$ concentrations measured by the WET Labs (USA) fluorometer did not differ significantly from





Chl-*a* concentrations determined from water samples (Meyer et al., 2017), no further calibration than the in-factory calibration was applied to the fluorometer data (see Loginova et al. (2016) for details). In order to obtain further subsurface hydrographic information, 53 temperature profiles organized in two sections across the front with a horizontal resolution of 0.3-0.5 km were measured using the CTD sensors mounted on a microstructure profiler. During these measurements the speed of the vessel was

reduced to about 0.75 m s$^{-1}$. The two sections are thereafter referred to as transects "B" and "C".

Surface diabatic heating was calculated from underway measurements as $Q_{net} = Q_{lw} + Q_{sw} + Q_{sh} + Q_{lh}$, i.e. the sum of longwave and shortwave radiation as well as sensible and latent heat fluxes. Net shortwave radiation into the ocean was estimated using underway measurements of downward shortwave radiation under consideration of the surface albedo effect (Payne, 1972). Net longwave radiation into the ocean was calculated as the difference between the measured downward long-

wave radiation and outgoing longwave radiation estimated using Stefan-Boltzman's law (with an emissivity of 0.985) applied to the thermosalinograph's temperature measurements. Latent and sensible heat fluxes along the cruise track were calculated using bulk formulae including the Webb-correction to the latent heat flux (Fairall et al., 1996b). During the experiment an average flux of latent heat of 20 W m$^{-2}$ *into* the ocean was observed. This coincided with high relative humidity and partially foggy conditions. Neither a cool skin nor a warm skin correction is applied to the thermosalinograph's temperature measure-

ments. A cool-skin would primarily form during night time, but is estimated to be on average less than 0.02 K cooler than the temperature measured by the thermosalinograph (Fairall et al., 1996a). A potentially larger warm-skin correction, which would increase outgoing longwave radiation mainly during periods of strong insolation, is not applied. This error in heatflux is however small compared to the dominant heatflux related to shortwave radiation during daytime.

The large scale sea surface temperature (SST) was retrieved from NASA's OceanColor project as daily satellite MODIS Aqua

und Terra SST data (http://oceancolor.gsfc.nasa.gov/). For the large scale wind field Advanced Scatterometer (ASCAT) wind data was taken from the Asia-Pacifc Data-Research Center at the University of Hawai'i (http://apdrc.soest.hawaii.edu/datadoc/ascat.php). The spatial resolution of SST was 4 km, while the wind data was available on a 0.25° grid. Nevertheless, it should be noted, that the ubiquitous presence of clouds during the main experiment period hindered an extensive use of remote sensing datasets.

## 2.2 Underway CO$_2$ measurements

CO$_2$ measurements were conducted by means of an underway system as described in Arévalo-Martínez et al. (2013). Seawater was drawn on board from a depth of about 6 m by means of a LOWARA submersible pump which was installed at the ship's moonpool. Atmospheric measurements were carried out every six hours by means of an AirCadet pump (Thermoscientific Inc., USA) which continuously brought air from 35 m height into the laboratory. Likewise, control measurements of standard gas mixtures with 201.0 and 602.4 ppm CO$_2$ were used in order to post-correct our data due to instrumental drift. The gas standards

were prepared at Deuste Steininger GmbH (Mühlhausen, Germany) and were calibrated at the Max Planck Institute for Bio-geochemistry (Jena, Germany) against the World Meteorological Organization standard scale. CO$_2$ data calibration as well as computation of CO$_2$ fugacities ($f$CO$_2$) was done according to the guidelines from Dickson et al. (2007). We report all seawater and atmospheric CO$_2$ values as 1-minute means. For this, we used a mean surface salinity from all CTD measurements of the M93 campaign (35.04), as salinity was not available from underway measurements. The air-sea flux of CO$_2$ was computed by





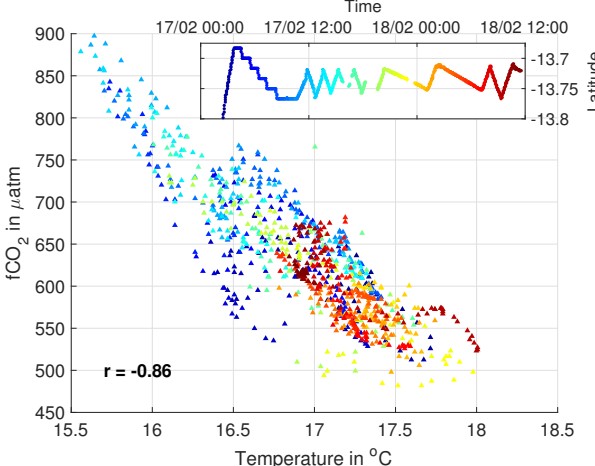

**Figure 3.** Correlation of surface temperature and $f\mathrm{CO_2}$ during the main experiment. The $f\mathrm{CO_2}$ values are already corrected for a 4 minute time lag. The correlation value $r$ is given in the bottom left corner. Data points are color-coded by their time of measurement. The top right box shows an orientation as to when the data was measured during the experiment. The color-coding carries the same information as the time axis.

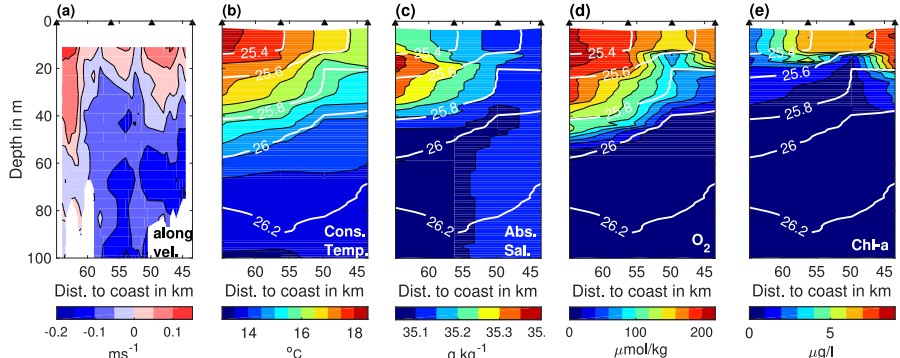

**Figure 4.** CTD Transect before the main experiment conducted on February 16 2013. Measured quantities contain along-frontal current velocities (a), conservative temperature (b), absolute salinity (c), $\mathrm{O_2}$ (d) and Chl-$a$ concentrations (e). Isopycnals are given by white lines in (b) - (e). Around CTD stations the along-frontal velocity is horizontally linearly interpolated.

using $F = kK_0(f\mathrm{CO_{2sw}} - f\mathrm{CO_{2air}})$, where $k$ is the air-sea gas exchange coefficient, calculated using the parameterization of Nightingale et al. (2000) with wind speeds standardized to $10\,\mathrm{m}$ height (Garratt, 1977). $K_0$ is the solubility of $\mathrm{CO_2}$ in seawater, calculated with the equations and coefficients from Weiss (1974) and Weiss and Price (1980), and $f\mathrm{CO_{2sw}}$ and $f\mathrm{CO_{2air}}$ are the fugacities of $\mathrm{CO_2}$ in seawater and atmosphere, respectively. The strongest correlation between the underway temperature and

5   $f\mathrm{CO_2}$ measurements was found at a time lag of 4 minutes ($r = -0.86$) (Fig. 3), which is probably due to the travel time for the seawater from the uptake to the underway $\mathrm{CO_2}$ sensor. Thus $\mathrm{CO_2}$ fugacities and fluxes are corrected for this time lag.





## 3 Physical and biogeochemical properties of the upwelling front

Near 14° S and 76°30' W the Peruvian upwelling region is characterized by a strong (quasi-) permanent upwelling cell (Fig. 1). The corresponding "upwelling front" was intensively sampled during the M93 cruise from February 16 to 18, 2013. The experiment's procedure was as follows: Prior to the main experiment (February 16 from 10:00 to 15:00), a CTD transect (black squares in Fig. 2) was conducted to document the horizontal circulation as well as the initial vertical distribution of temperature, salinity, $O_2$ and Chl-$a$ across the front (Fig. 4). Starting on February 17 at 04:00, the upwelling front was mapped by seven $\sim$10 km long zonal transects. These were conducted from north to south in 1.8 km spacing (Fig. 2). Each zonal transect took about 45 minutes. From the highest and lowest surface temperature recorded, a cross-frontal axis was estimated (dashed line in Fig. 2). Subsequently, 17 cross-frontal transects were conducted along this axis to study the variability of the front on timescales of several hours (Fig. 5). Among these, the high-resolution temperature sections B and C were conducted as cross-frontal transects 12 and 14. The high resolution transects took 4.5 hours each to complete, while a regular cross-frontal transect was completed in 40 minutes.

The experiment period can be divided into three different wind forcing regimes. The first regime is characterized by down-frontal winds with speeds of over 10 m s$^{-1}$ (Fig. 6). It lasts from the beginning of the experiment including the CTD transect until February 17 11:00, i.e. right after the beginning of the cross-frontal transects. During this phase, the front is characterized by a strong cross-frontal surface temperature gradient of about 1°C over 10 km (Fig. 5a). The CTD transect reveals the strong subsurface frontal signature in the temperature and salinity field (Fig. 4b,c). The mixed layer exhibits almost vertical isopycnals and is estimated to be about 15 m deep, using a $\Delta T = 0.2$°C criterion. The southward flowing Peru Chile Undercurrent (PCUC) weakens from $\sim$ 20 cm s$^{-1}$ at 80 meters depth towards the surface (Fig. 4a). The along-frontal velocity is low in the shallowest ADCP bin throughout the experiment (Fig. 7). As a result, lateral along-frontal advection is likely to play a minor role at the surface. In agreement with extensive upwelling of cold, nutrient rich waters during the strong wind period the Chl-$a$ concentrations on the onshore side of the front were strongly enhanced with concentrations reaching 7 $\mu$g l$^{-1}$ in the mixed layer (Fig. 4e). Simultaneously, the subsurface $O_2$ minimum was drawn closer to the surface on the onshore side of the front (Fig. 4d). Concentrations below 100 $\mu$mol kg$^{-1}$ could be found at 15 m depth and values dropped below 20 $\mu$mol kg$^{-1}$ already beneath $\sim$ 40 m. Further, a gradient in surface $f$CO$_2$ was discernible during the CTD transect (not shown) and during the zonal ship sections across the front (Fig. 2b), increasing from 600 $\mu$atm in offshore locations up to over 800 $\mu$atm on the onshore side of the front. During this period, peak sea-to-air CO$_2$ fluxes of over 80 mmol m$^{-2}$ d$^{-1}$ are measured on the cold side of the front (Fig. 5d).

## 4 Cross-frontal changes after weakening of downfrontal winds

During the second wind period from 11:00 to 21:00 on February 17 the wind continues to blow in the downfrontal direction but drops to almost 5 m s$^{-1}$ (Fig. 6). During this period the clear surface temperature gradient signal is disturbed by coherent anomalies appearing at 50 km off the coast (Fig. 5b). These anomalies grow in amplitude and lead to a break up of the clear temperature gradient at around 20:00. At the same time the strong outgassing of CO$_2$ is reduced on the onshore side of the



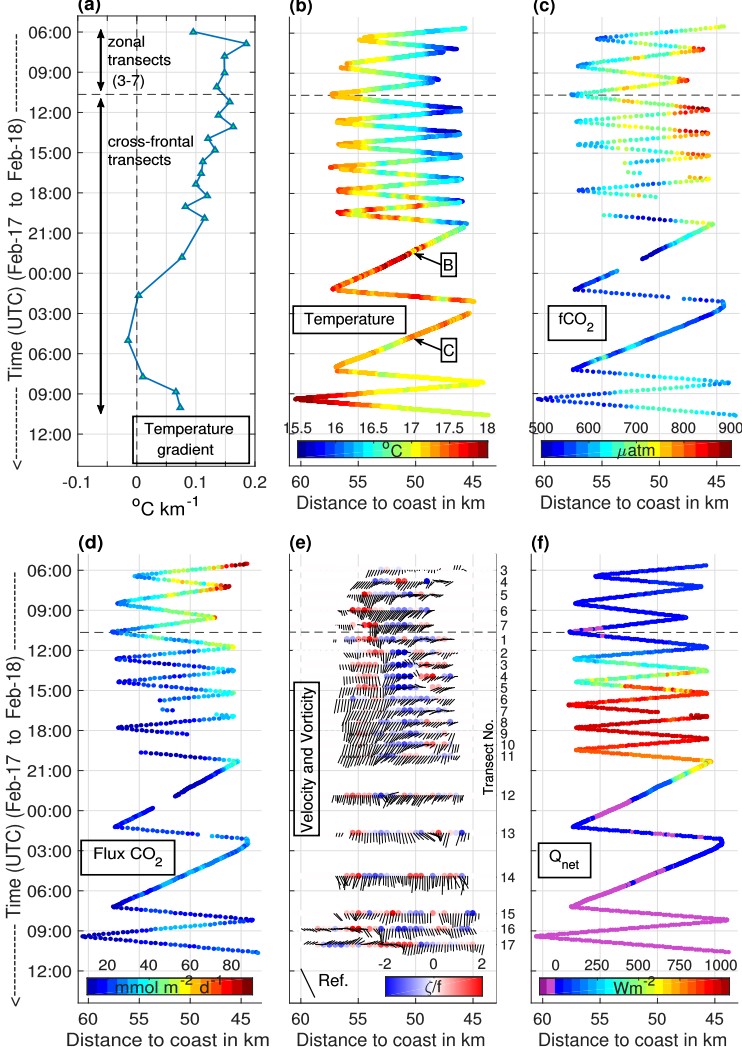

**Figure 5.** Hovmoeller diagrams of the cross-frontal surface temperature gradient (a), underway surface temperatures (b), surface $f\mathrm{CO_2}$ (c), ocean-atmosphere $\mathrm{CO_2}$ fluxes (d), mean current velocities in the upper 40 m and vorticity (e) and diabatic surface heating (f) for the last five zonal transects and the subsequent 17 cross-frontal transects. The horizontal dashed line indicates the transition from zonal transects to repeated cross-frontal transects. Temperature gradients in (a) are calculated using a linear fit to the surface temperature transect shown in (b). In (b) the position of transects B and C is indicated. The vorticity of the vertically averaged velocity in (e) is calculated as $\zeta = \partial u_{al}/\partial y$ (Rudnick, 2001), where $u_{al}$ is the along-front velocity and $y$ is the cross-frontal distance, taken to be 2 km, comparable to the mixed layer deformation radius. The reference arrow in the bottom left corner anchored at 60 km from the coast indicates a flow with $0.3\ \mathrm{m\ s^{-1}}$ in the onshore (cross-front) and $0.3\ \mathrm{m\ s^{-1}}$ in the southward along-front direction. The vorticity is normalized by the Coriolis parameter to give a proxy for the Rossby number. Velocity and vorticity are plotted at the time mean for each transect. Time is given in UTC.



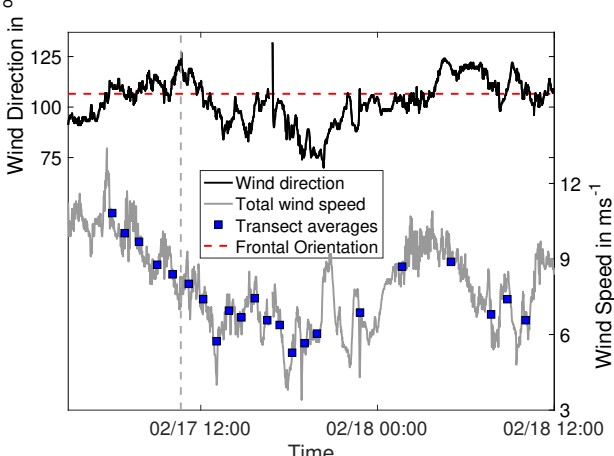

**Figure 6.** Time series of total wind speed (grey) and wind direction (black) from underway measurements at 35.3 m height. Blue rectangles shown mean total wind speed for each zonal and cross-frontal transect. The dashed red line shows the (initial) frontal orientation deduced from the frontal mapping. The vertical dashed grey line indicates the transition from zonal to cross-frontal transects. Time is given in UTC.

front (Fig. 5d), while surface $f\mathrm{CO_2}$ values remain rather high until about 18:00 (Fig. 5c). At the beginning of the transition to the low wind speed period, a strong offshore directed velocity signal is observed close to the surface on the onshore side of the front (Fig. 7a). The velocity maximum of up to 15 $\mathrm{cm\ s^{-1}}$ deepens with increasing distance from shore. During the wind minimum, maximum offshore velocities are found at 25-30 m depth, exceeding 20 $\mathrm{cm\ s^{-1}}$ (Fig. 7c). At the same time, the

along-frontal velocity field exhibits a slight reduction in vertical shear (Fig. 7b,d).

At about 21:00 on February 17 the along-frontal wind begins to increase again and reaches about 9 $\mathrm{m\ s^{-1}}$, defining the beginning of the third wind regime. Initially, the surface temperature gradient vanishes and even reverses slightly, before strengthening again towards February 18 09:00. Both high-resolution temperature transects (B and C) exhibit an average mixed layer depth of 3 and 5 m respectively (Fig. 8). The sections show that temperature changes are not limited to the surface

layer but reach about 40-50 m deep. While the temperature field in transect B exhibits a clear frontal structure, the isotherms are close to horizontal in transect C, with exception of undulations associated with internal waves propagating onshore (e.g. at km 46 in Fig. 8b). As a result, the mean cross-frontal velocity field shows no longer any clear offshore directed velocity signal (Fig. 7e). Also the subsurface along-frontal velocity maximum of the PCUC appears to be weaker (Fig. 7f). During this last phase the outgassing of $\mathrm{CO_2}$ is at its minimum with fluxes less than 20 $\mathrm{mmol\ m^{-2}\ d^{-1}}$ despite the again increased wind

speeds (Fig. 5d). $f\mathrm{CO_2}$ remains at ~600 $\mu\mathrm{atm}$ during this phase and increases only slightly along the last two transects during which the front starts to reform.

Figure 3 shows the correlation of surface temperature and $f\mathrm{CO_2}$ during the main experiment. The high negative correlation indicates the governing effect of temperature on the $\mathrm{CO_2}$ field. Especially during the beginning of the experiment the high $f\mathrm{CO_2}$ values are only found on the cold side of the front. Towards the end of the experiment the clear correlation signal breaks

up. The changes in temperature and $f\mathrm{CO_2}$ are observed over the course of $\mathcal{O}(\text{hours})$, giving phytoplankton only a short time





window to react. It is thus likely that physical processes are mainly responsible for the measured variability in surface $f$CO$_2$. In the following part of this study, we analyze the observed changes across the upwelling front in the context of physical processes that could be responsible for driving the observed variability.

### 4.1 Surface heat fluxes

The changes across the front are observed within a 24 hour period. The wind begins to drop at 11:00 on February 17 (06:00 LT) and the temperature gradient fully vanishes 14 hours later at 01:00 on February 18 (Fig. 5a). As this time period matches well with the diurnal cycle of solar insolation, it could be hypothesized that differential heating throughout the day causes the reduction of the cross-frontal temperature gradient. Figure 5f shows the diurnal cycle of net surface heat flux. During the zonal transects the ocean surface is heated on average by 50 W m$^{-2}$. In the absence of solar insolation, this heatflux is mostly

maintained by latent and sensible heat fluxes *into* the ocean, overcoming the ocean's weak heat loss due to outgoing long wave radiation (individual heat flux components not shown). At around 12:00 the net heat flux into the ocean begins to strengthen, associated with an intensification of solar insolation. During the fourth and the beginning of the fifth cross-frontal transect the heat flux into the ocean is hampered due to irregularities in the incoming shortwave radiation. The net heat flux reaches its peak of 1010 W m$^{-2}$ at 18:00 and reduces subsequently. From 23:00 onwards, the net heat flux fluctuates around 0 W

m$^{-2}$. During the last cross-frontal transects the surface is continuously cooled by less than 50 W m$^{-2}$. During the phase of strongest heating, the heat flux shows only small lateral differences of mostly less than 50 W m$^{-2}$ across the temperature front. Further, no clear pattern of slightly stronger heating on the cold side of the front is discernible. As a result, a uniform warming of the surface waters is expected. Sectional averaging of the first 13 cross-frontal transects, corresponding to the time period of incoming solar radiation, shows an increase of the transects' mean temperature of 0.8°C. From $\Delta Q = \rho_r c_p V \Delta T$

with the reference density $\rho_r = 1035$ kgm$^{-3}$ and a heat capacity $c_p = 4000$ Jkg$^{-1}$ m$^{-3}$ it can be roughly estimated that over the same time period of 14.5 hours the average net heat flux causes a warming of a surface water column of volume $V$ with unit area and 10 m depth by 0.6°C. Thus, the background temperature increase is most likely linked to the surface heat fluxes. The small scale temperature anomalies develop during the phase of strong heating (Fig. 5b). Using the same calculation as above the maximum heatflux of 1010 W m$^{-2}$ can warm the surface water by almost 0.1°C per hour. Even though this roughly

corresponds to the amplitude of the temperature anomalies the lateral homogeneity of the net surface heat flux makes it difficult to attribute the development of the anomalies to surface heating.

     The temperature front finally vanishes during a phase of almost no net surface heat flux. The high-resolution temperature transects B and C (Fig. 8) show, that during this phase, cooling takes place on the offshore side of the front, while the onshore side is significantly warmed over a period of 6 hours. On the onshore side, temperature changes of $\sim 2$°C occur up to $\sim 40$

m depth. A heatflux of $\sim 3800$ W m$^{-2}$ is required to heat a water column of 40 m by 0.5°C using the rough estimate above. Likewise, a comparably strong cooling is required on the offshore side of the front in order to explain the observed temperature changes. Neither the two heatfluxes nor the associated strong cross-frontal gradient are within the range of the observed values.

     The net surface heat flux can be converted into a vertical buoyancy flux following $\langle w'b' \rangle_{Q_{net}} = \alpha g Q_{net}/(\rho c_p)$, where $w'$ is the vertical velocity anomaly, $b'$ is the buoyancy anomaly, $g$ is the gravitational acceleration and $\alpha$ is the thermal expansion





coefficient. Due to the lack of data, the calculation of the surface density $\rho$ requires an assumption about the salinity. The inital CTD section shows a surface salinity varying between 35.05 and 35.2 (Fig. 4c). Thus, the surface water density $\rho$ is calculated using a constant salinity of 35.1, yielding an uncertainty of density in the range of $0.1 \ \mathrm{kg \ m^{-3}}$ and in vertical buoyancy fluxes of 1%. The mean values of $\langle w'b' \rangle_{Q_{net}}$ for each transect are shown in Fig. 9. During both, night and day time, the surface layer

gains buoyancy. In the night this vertical buoyancy flux is weak, with a rate of less than $1 \times 10^{-8} \ \mathrm{m^2 \ s^{-3}}$. During the day, vertical buoyancy fluxes of more than $5 \times 10^{-7} \ \mathrm{m^2 \ s^{-3}}$ act to stratify the surface layers.

While surface heat fluxes are shown to contribute to the overall evolution of the temperature field, the observed heat fluxes do not yield the required strong gradients across the front and on even shorter lengthscales. Especially, the observed temperature changes below the surface layer seem to be unconnected to surface heating. Thus, dynamical processes are investigated in the

following to understand the reduction of the cross-frontal temperature gradient.

## 4.2 Ekman transport

Throughout the experiment the wind speed changed significantly, while the wind direction stayed almost parallel to the front in the same direction as the frontal jet, i.e. downfrontal (Fig. 6). The cross-frontal Ekman velocity is calculated as $u_{Ek} = \tau_0/(f \rho_0 H_{ML})$, where $\tau_0 = \rho_{air} \left( c_d u_{10}^2 \right)$ is the neutral along-frontal surface wind stress. The drag coefficient $c_d$ and the along-

frontal wind speed at 10 m height $u_{10}$ are calculated following Smith (1988). The mean density within the Ekman layer $\rho_0$ is approximated by the surface density $\rho$. Assuming a mixed layer depth $H_{ML} = 10$ m Ekman current velocities reach up to 0.5 $\mathrm{m \ s^{-1}}$ and are always directed offshore (figure not shown). During the period of the developing surface temperature anomalies, the Ekman velocity drops to $\sim 0.1 \ \mathrm{m \ s^{-1}}$, agreeing with the observed cross-front velocities close to the surface (Fig. 7a,c). The large uncertainties in surface salinity lead to only minor errors in Ekman velocity of less than $10^{-3} \ \mathrm{m \ s^{-1}}$. The choice of

the mixed layer depth has a far larger impact due to the inverse proportionality. For example choosing $H_{ML} = 5$ m leads to Ekman velocities larger by a factor of 2. The clear offshore direction of the Ekman transport is however not put into question by these uncertainties.

The wind driven Ekman transport is associated with a non-geostrophic overturning circulation in the vertical/cross-frontal plane (Thomas et al., 2008). For such an overturning circulation it is possible to infer the associated vertical buoyancy flux

$\langle w'b' \rangle$. The Ekman buoyancy flux (EBF) is given by $\langle w'b' \rangle_{EBF} = \tau_0 \cdot \nabla b/(\rho_0 f)$ (Thomas and Lee, 2005), where $f$ is the Coriolis parameter. $\nabla b$ is the cross-frontal buoyancy gradient, taken as the slope from a linear fit to the surface buoyancy across the front. Buoyancy is calculated as $b = -g(\rho - \rho_r)/\rho_r$ using a reference density $\rho_r = 1035 \ \mathrm{kg \ m^{-3}}$.

Calculating the buoyancy gradient with a constant salinity value of $35.1 \ \mathrm{g \ kg^{-1}}$ is likely to yield an error for the Ekman buoyancy fluxes as the initial CTD transect indicates a cross-fontal surface salinity gradient of $-5 \times 10^{-3} \ \mathrm{g \ kg^{-1} \ km^{-1}}$. Hence,

this salinity gradient is imposed onto the mean value of $35.1 \ \mathrm{g \ kg^{-1}}$ to estimate the cross-frontal buoyancy gradient. Figure 9 shows the buoyancy fluxes calculated for each transect. The EBF is enclosed by an uncertainty range related to the surface salinity gradient. The edges of this uncertainty range stem from the case of no cross-frontal salinity gradient and of double the initially observed cross-frontal gradient.



The downfrontal wind drives a continuous Ekman transport from the cold to the warm side of the front, thus acting to keep the isopycnals strongly tilted, rather than directly transporting warm water onshore. Correspondingly, the EBF is predominantly negative, signifying a destratification within the surface layer, inhibiting the slumping of isopycnals. During the cross-frontal transects, the EBF weakens drastically from $-1 \times 10^{-6}$ m$^2$ s$^{-3}$ to $-2 \times 10^{-7}$ m$^2$ s$^{-3}$. Only during cross-frontal transects 13-15 the buoyancy flux changes sign, which goes along with the turnaround of the surface temperature gradient (Fig. 5a) while the Ekman velocity continues to be directed offshore. Afterwards, during the reformation of the front towards the end of the experiment on February 18, the EBF returns to negative values.

The weakening of the temperature front is not caused by onshore Ekman transport. Still, the reduction in downfrontal wind speed and the associated weaking in offshore Ekman transport can change the frontal stability. During most of the experiment's duration, the EBF inhibited a slumping of isopynals. However, this flux weakened over the course of the experiment, potentially allowing other processes such as surface heat fluxes or mixed layer instabilities to become more important.

### 4.3 Mixed layer instabilities (MLIs)

Baroclinic instabilities confined to the mixed layer are referred to as submesoscale mixed layer instabilities (MLI) (Haine and Marshall, 1998; Boccaletti et al., 2007). They act to restratify the mixed layer by extracting potential energy from horizontal density gradients within the vertically well mixed surface layer and convert this into eddy kinetic energy, by formation of mixed layer eddies which accomplish the cross-frontal transport and exchange. With their short timescales of $\mathcal{O}(f^{-1})$ submesoscale MLI are thought to be efficient in converting lateral density fronts to a stratified mixed layer. Their horizontal scales are given by $L = NH_{ML}/f$, where $N = \sqrt{g/\rho (\partial\rho/\partial z)}$ is the small but non-vanishing stratification within the mixed layer (Thomas et al., 2008). Based on the initial CTD transect A (shown in Fig. 4) this lengthscale $L$ is calculated to be $2.5\pm1.5$ km. The large error is accounting for the uncertainties in the mean mixed layer stratification, which is estimated to be $N \approx (6\pm4) \times 10^{-3}$ s$^{-1}$. The inverse of the growth rate of MLI ($2\pi/f$) can be used as the associated time scale which amounts to $\sim 2.1$ days in the present case. The proximity of the study region to the equator associated with relatively long inertial timescales, makes it possible to capture well the variability by means of underway measurements.

Repeated observations of coherent surface temperature anomalies and their intensification between 15:00 and 20:00 on February 17 around 50 kilometers away from the coast might indicate the development of MLI (Fig. 5b). Preceeding their appearance are small anomalies in the current field averaged over the upper 40 m (Fig. 5e). There, the mostly southbound flow is reversing and weakly flowing northward, producing large relative vorticity values (cross-frontal transects 3, 4, 5). This points towards eddying structures which could be associated with MLI. The mixed layer depth, estimated by a $\Delta T = 0.2°$C criterion, shows a shallowing throughout the experiment. While the average mixed layer depth is 15 m during the CTD transect A it reduces to 3 and 5 m during transects B and C, respectively. We applied Linear Stability Analysis (LSA), which provides vertical modes of the growth of baroclinic instabilities (Brüggemann and Eden, 2014; Thomsen et al., 2014) to the laterally averaged stratification and geostrophic current shear profile from the intial CTD section (Fig. 10b,c). This analysis reveals the existance of both, a deep mesoscale mode and a shallow mode, confined to the mixed layer (Fig. 10a,d,e). The horizontal length scale of the deep mode is about 200 km, while it is only 8.7 km for the shallow mode. This length scale is of the same order of



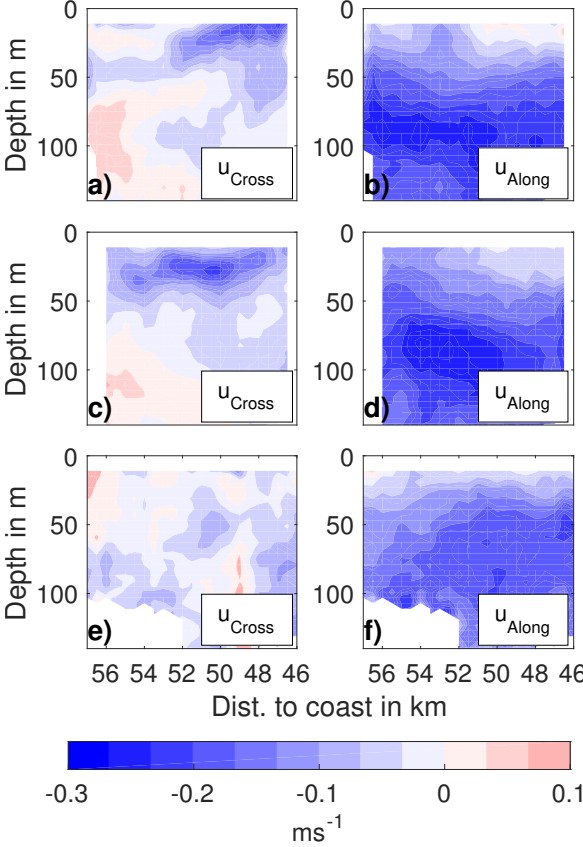

**Figure 7.** Mean cross- and along-front current velocities for cross-frontal trasects 1-6 in (a) and (b), 7-11 in (c) and (d) and 12-17 in (e) and (f). Currents are horizonatally binned onto a 500 m grid and subsequently averaged.

magnitude as the expected mixed layer deformation radius $L$ and the size of the observed surface temperature anomalies. The initial growth rate of the shallow mode calculated from LSA is $0.5 \ \mathrm{day}^{-1}$. Once the instabilities grow too large, nonlinearities dominate the instability process and LSA may not provide a useful description of the evolution of the instabilities (Thomsen et al., 2014). However, the calculated growth rate is relatively slow compared to the observed rapid decline of the cross-frontal

5 temperature gradient. Also the LSA-inferred shallow mode is limited to $\sim 15$ m depth, i.e. the mixed layer (Fig. 10e). Thus, instability-induced changes of the velocity field at 30 m depth would require an interaction with the deep mode (Fig. 10d), which according to LSA shows no large cross-front velocity anomalies.

Similar to Ekman dynamics, a secondary overturning circulation is driven by MLI. It is important to note, that while an EBF is associated with strong diapycnal mixing, pure mixed layer instabilities are of an adiabatic nature. According to Fox-

10 Kemper et al. (2008) the vertical buoyancy flux due to MLI given by $\langle w'b' \rangle_{MLI} = CH^2_{ML}(\nabla b)^2/|f|$ (where $C = 0.08$ is a



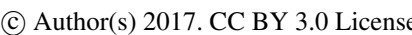


**Figure 8.** Temperature (black contour lines) and cross-front velocity (color-coded) for Transects B (a) and C (b) conducted as the 12th and 14th cross-frontal transect, starting on February 17 21:00 and February 18 03:00, respectively. Black crosses show the location of the temperature profiles from the microstructure probe. For both transects the ship moved away from the coast (right to left).

constant) always tends to reduce the lateral temperature gradient. Throughout the experiment the $\langle w'b' \rangle_{MLI}$ values are positive, indicating a restratification of the mixed layer. However, the buoyancy flux associated with MLI is in general about 10 times smaller than the EBF. During the stable front the buoyancy flux associated with MLI $\langle w'b' \rangle_{MLI}$ is less than $1 \times 10^{-7}$ m$^2$ s$^{-3}$ and reduces afterwards. Calculations are done with $H_{ML} = 15$ m based on the CTD transect. This is likely to be an upper

bound for the mixed layer depth during the main experiment, as transects B and C show a much shallower mixed layer. Both LSA and buoyancy flux estimates imply that MLI are present, acting to restratify the mixed layer, but seem to be too slow and too confined to the surface layer, such that observed changes at depth cannot be explained by MLI alone.

### 4.4 Surface gravity current

In the presence of lateral density gradients, an unforced surface mixed layer is subject to pressure driven gravity currents,

where lighter water spreads on top of the denser waters on timescales below the inertial period, i.e. independent of rotational effects (e.g. Kao et al. (1977)). The observed reduction in downfrontal wind speed could therefore give rise to the spreading of a buoyant plume down the temperature gradient. The rapid decline and even a slight reversal of the temperature front is captured well by transects B and C (Fig. 8). Within 6 hours the structure of both the temperature and cross-frontal velocity fields changed significantly. While transect B still shows a pronounced (subsurface) frontal signature, the isotherms in transect

C are close to being horizontal. The temperature field in transect C is subject to distortion through internal waves. At km 46, the undulating isotherms go along with an oscillating signal in cross-front velocity, agreeing with internal waves propagating up the continental shelf. The strong depression of isotherms at km 57 could be the onset of another internal wave signal. In transect B, the temperature and velocity field exhibit a strong anomaly at km 52.5. There, a narrow but strong depression of





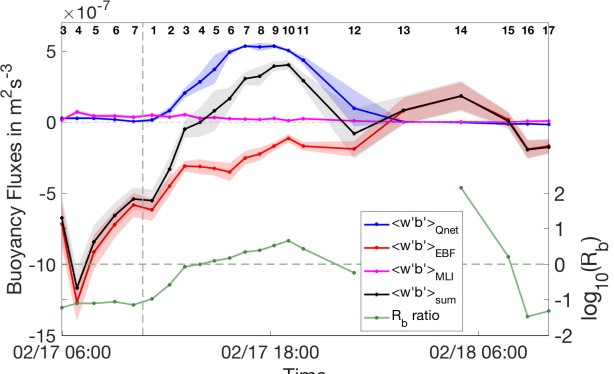

**Figure 9.** Vertical buoyancy fluxes associated with surface heating (blue), Ekman transport (red) and mixed layer instabilities (magenta). The uncertainty range for Ekman buoyancy fluxes (EBF) and buoyancy fluxes associated with mixed layer instabilities (MLI) is accounting for uncertainties in the surface salinity gradient (see text for more details). The uncertainty range for vertical buoyancy fluxes due to surface heating is given as one standard deviation obtained from sectional averaging. The sum of the three processes is given by the black line. For the $R_b$ ratio the errors are omitted. $\log_{10}(R_b) > 0$ points towards a net restratification, while $\log_{10}(R_b) < 0$ indicates a destabilization of the water column. For calculation details of all quantities see text. Time is given in UTC. The small black numbers on top indicate the respective zonal or cross-frontal transect number.

the temperature field coincides with a region of strong convergence in the cross-frontal direction. Unlike in transect C, the attribution of the depression of the isotherms to an internal wave signal is not straightforward as the velocity field does not show an oscillating behaviour in the vicinity of the depression. However, this might be attributed to aliasing of the signal due to too few hydrographic observations, which do not capture the short period of the internal waves.

Any temperature front is associated with a cross-frontal pressure gradient. In coastal upwelling regimes, this pressure gradient is largely in geostrophic balance manifesting itself in an along-frontal jet. However, if the wind field setting up the frontal system becomes too weak, the tilted isopycnals might start to slump, with the light surface waters offshore pushing over the denser surface water located further onshore. Thus, baroclinic temperature fronts can be eroded by gravity currents flowing down the pressure gradient across the front. Observations from river plumes show that the head of a gravity current may excite

large-amplitude internal waves (Nash and Moum, 2005). These are either arrested at the leading edge of the gravity current or may propagate ahead of it if the current's advection speed is lower than that of the wave propagation speed. Assuming that the anomaly signals in transect B (at km 52.5) and C (at km 46) belong to the same internal wave package that propagates at the leading edge of the gravity current, a propagation speed of $0.4\,\mathrm{m\,s^{-1}}$ is estimated using a time difference of 4.5 hours and a distance of $6.5\,\mathrm{km}$. In a simple 2-layer model, the propagation speed of the gravity current in deep ambient fluid is given

by $c = \sqrt{g'H}$, where $g' = g\Delta\rho/\rho$ is the reduced gravity, $\Delta\rho$ is the density difference between the two layers and $H$ is the thickness of the upper layer (Shin et al., 2004; Dale et al., 2008). Assuming a 2-layer problem in the observed frontal setup with $H = 30\,\mathrm{m}$, as the average depth of the 15°C isotherm, and $g' = 5 \times 10^{-3}\,\mathrm{m\,s^{-2}}$ derived from densities calculated at 10 and 50 m depth, assuming a salinity of 35, the theoretical propagation speed $c$ can be estimated to be $0.4\,\mathrm{m\,s^{-1}}$, thus agreeing


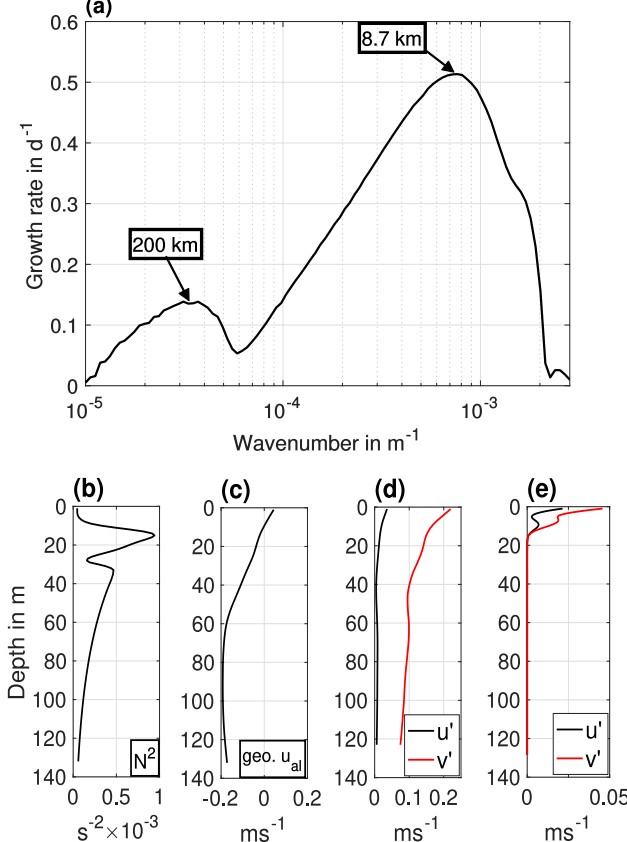

**Figure 10.** (a) Initial growth rate of barolinic instabilities obtained from linear stability analysis, applied to the frontal state during the CTD section (transect A). The smoothed stratification and geostrophic along-frontal velcity profiles used as input, are shown in panels (b) and (c). A deep mode at large horizontal scales and a shallow mode, associated with mixed layer instabilities and short horizontal length scales are present. The corresponding anomalies for the cross-front (u) and along-front (v) velocities are shown in panels (d) and (e) for the deep and shallow modes, respectively.

with the observed propagation speed. If the temperature anomalies in sections B and C represent an internal wave package propagating at the leading edge of a gravity current, the internal wave propagation speed estimates above qualify a density driven gravity current as a possible mechanism behind the abrupt degradation of the temperature front. In contrast to the other analyzed processes it is capable of explaining subsurface changes in the temperature field.





## 5   Comparison of buoyancy fluxes

Over the whole experiment, three of the four analyzed physical processes are comparable in terms of the associated vertical buoyancy fluxes (Fig. 9). The ratio

$$R_b = \frac{\langle w'b' \rangle_{Qnet>0} + \langle w'b' \rangle_{MLI} + \langle w'b' \rangle_{EBF>0}}{\langle w'b' \rangle_{Qnet<0} + \langle w'b' \rangle_{EBF<0}} \tag{1}$$

allows for a statement on the combined effect of the Ekman buoyancy flux, surface heating and MLI (Mahadevan et al., 2010; Taylor and Ferrari, 2011). The numerator only contains terms associated with stratification, i.e. positive buoyancy fluxes. For example the buoyancy flux associated with MLIs is always positive and thus appears only in the numerator. The EBF however changes sign during the experiment. During the parts of the experiment with positive EBF it will contribute to the term $\langle w'b' \rangle_{EBF>0}$ in the numerator. If the EBF is negative, its modulus will appear in the denominator as $\langle w'b' \rangle_{EBF<0}$.

The same applies to the surface heating buoyancy fluxes. As a result, if $R_b > 1$, the mixed layer is subject to restratification associated with a slumping of isopycnals, whereas $R_b < 1$ implies a destratification, i.e. mixing in the surface layer. During the time of the developmet of the coherent surface temperature anomalies (cross-frontal transects 7-11), the combination of the three processes favor a restratification of the mixed layer (Fig. 9). Linear stability analysis confirms the existance of a shallow baroclinic mode, which shows growth rates of $0.5 \, \mathrm{d}^{-1}$ on lengthscales of $8.7 \, \mathrm{km}$. The latter agrees well with the extent of the

observed temperature anomalies and the mixed layer radius of deformation calculated from the initial CTD transect. While the temporal scale of the shallow mode is in agreement with the development of the surface temperature anomalies, its impact in the rapid decline of the temperature front around 00:00 on February 18 is probably low, as the growth rate at the initial phase of the instability seems too small. Further, the low associated buoyancy fluxes (Fig. 9) indicate that MLI may contribute only to a small degree to the observed variability across the front.

In contrast, the buoyancy fluxes associated with surface heating and cross-frontal Ekman transport (Fig. 9) contribute to the changes in sign of the sum of the buoyancy flux (and to $R_b > 1$) during February 17. Even though the strength of the diabatic heating agrees well with the amplitude of the developing surface temperature anomaly, stronger than observed horizontal gradients in surface heating are required to induce the observed lateral differences in warming. Further, the nature of this buoyancy flux is purely vertical and thus unable to create horizontal gradients in the case of spatially uniform heating, opposed

to buoyancy fluxes associated with MLI or Ekman dynamics, which are fully 3-dimensional.

The buoyancy flux related to a gravity current is mainly lateral. However, if the warm water spreads on top of the cold water on the onshore side of the front, a vertical buoyancy flux is induced. In a closed domain the mean vertical buoyancy flux can be estimated following $\partial E_p / \partial t \approx -\langle wb \rangle$ (Peltier and Caulfield, 2003; White and Helfrich, 2013), thus by comparing volume-averaged potential energies $E_p$ at two different points in time $t$. In a 2-dimensional domain the volume-averaged potential

energy is calculated as $E_p = (hd)^{-1} \int_d \int_0^h bz \, dz \, dx$, where $d$ is the lateral distance and $h$ is depth. Using a constant salinity value of 35 the buoyancy field is calculated for transects B and C. Considering the upper $70 \, \mathrm{m}$ on the initially cold side of the front (i.e. only east of 52 km offshore) a vertical buoyancy flux of $2.7 \times 10^{-6} \, \mathrm{m}^2 \, \mathrm{s}^{-3}$ is estimated. Laterally extending the domain to the full transects B and C reduces the vertical buoyancy flux to $1.8 \times 10^{-6} \, \mathrm{m}^2 \, \mathrm{s}^{-3}$. As the transects B and C have no closed boundaries, the estimates carry large uncertainties. The pressure driven gravity current is however assumed




to be fully captured by both transects so that its associated effect should be captured. In the context of a closed domain the subsurface temperature reduction on the initialy warm side of the front could be associated with an upward suction of colder water balancing the downward pumping of surface water on the cold side of the front (Fig. 8).

The estimated flux due to the gravity current is much larger than the buoyancy fluxes associated with the other above
analyzed processes during the cross-frontal transects 12-14, regardless of the chosen cross-frontal width considered. Only the EBF during the beginning of the experiment is of the same order of magnitude.

## 6   Discussion

Gathering data from multiple ship surveys Friederich et al. (2008) observed oceanic outgassing of $CO_2$ in the Peruvian upwelling region throughout the year. On average the authors estimated a sea-to-air $CO_2$ flux of $5 \, \mathrm{mol \, m^{-2} \, yr^{-1}}$. On seasonal
timescales, the flux varied between 2.5 and $10 \, \mathrm{mol \, m^{-2} \, yr^{-1}}$ associated with weak and strong upwelling periods, respectively. Although Friederich et al. (2008) focused on the large scale distribution of surface $CO_2$ fluxes and the associated mechanisms, they further detected strong variability of $CO_2$ fluxes on short time and space scales ($\mathcal{O}$(hours-days) and $\mathcal{O}$(km)), which contributes significantly to the overall $CO_2$ budget.

Motivated by their findings, the study presented here focuses on the pronounced submesoscale variability of surface $CO_2$
in the Peruvian upwelling region at 13.7° S. Our observations show $CO_2$ outgassing rates between 3.5 and $30 \, \mathrm{mol \, m^{-2} \, yr^{-1}}$ which are in line with the outgassing signals observed by Friederich et al. (2008), even after accounting for the difference in converting our values from partial pressures to fugacities. Our peak sea-air fluxes reaching up to $80 \, \mathrm{mmol \, m^{-2} \, d^{-1}}$ are distinctly higher than the maximum flux of $12.4 \, \mathrm{mmol \, m^{-2} \, d^{-1}}$ reported by Friederich et al. (2008) for a measurement campaign in the month of February. At the same time, our peak-flux is only slightly higher than maximum flux values reported
from other months. It should be pointed out, however, that the results from Friederich et al. (2008) are re-scaled to a much larger area (5° S - 15° S), have been subject of spatial extrapolation and smoothing, and were calculated with a sea-air gas exchange parameterization which tends to overestimate the gas transfer velocities (see Wanninkhof (2014)). Hence, it is difficult to draw a direct comparison between both data sets. Nontheless, both studies agree in that the near-coastal zone off Peru acts as a rather strong source of $CO_2$ to the atmosphere, and from our data it seems clear that the onshore side of the upwelling front could be
associated with an important enhancement of $CO_2$ outgassing. We analyzed a suite of processes involved in the evolution of the upwelling front and surface $f\mathrm{CO_2}$ on short time and space scales ($\mathcal{O}$(1) km, $\mathcal{O}$(0.5) day). The following conclusions can be drawn from the results:

The downfrontal wind is an important ingredient in maintaining tilted isopcynals in the surface layer. The EBF dominates all other buoyancy fluxes during the strong wind period (Fig. 9). In the study by Dale et al. (2008) a restratification of the
mixed layer is observed in connection with a frontal decay in the upwelling system off Oregon on length scales comparable to those presented here. In their study, a reversal of the wind direction plays a crucial role in driving the frontal decay by inducing an Ekman transport down the cross-frontal pressure gradient. For the frontal decay presented here, the wind is always directed down-frontal and does not change significantly. The cross-frontal wind component is weak (mostly less than $3 \, \mathrm{m \, s^{-1}}$)





and alternates in direction between onshore and offshore without any clear dominance (Fig. 6). As a result, Ekman transport and the associated buoyancy flux do not change direction. Still, the strong reduction in cross-frontal Ekman transport could potentially give way for other mechanisms.

Pronounced changes across the front are shown to occur in two steps after the distinct reduction in downfrontal wind speed

with a time lag of about 9 hours. The first step is characterized by the gradual development and strengthening of coherent surface temperature anomalies, while the second is characterized by a sudden decline and even slight reversal of the temperaure gradient. Transects B and C (Fig. 8) show that changes in the temperature field are thereby not limited to the surface layer, but reach down to 50 m. To identify the underlying processes, the frontal evolution is described above in a 2-dimensional framework, i.e. in the vertical/cross-frontal plane. The variability induced by along-frontal advection is neglected. This assumption

seems valid, as the along-frontal current velocities reduce close to zero in proximity to the surface (Fig. 7b,d,e). Still, the two dimensional framework allows for various hypotheses about the driving mechanisms behind the observed changes.

The analyses have shown that the diurnal surface heating is able to explain the majority of the mean increase in surface temperatures. During the phase of maximum heating the associated positive buoyancy flux into the ocean outweighs the permanent reduction in surface buoyancy by the down-frontal wind stress (Fig. 9). However, the spatially homogeneous heating is unable

to account for the developing small-scale temperatures anomalies. As a result, submesoscale MLI have been investigated as they are shown to restratify the surface mixed layer and are thereby capable of generating lateral gradients and anomalies. LSA indeed shows the presence of a shallow baroclinic mode confined to the mixed layer (Fig. 10). While the length scale of this mode roughly meets the observed horizontal extent of the temperature anomalies and the mixed layer deformation radius, the instability seems to grow too slowly to be the dominant dynamical process involved in the frontal decay. Hence, also the

vertical buoyancy fluxes asociated with MLI are an order of magnitude smaller than those related to surface heating or Ekman dynamics (Fig. 9).

The low-latitude location of the experiment site at $13.77°$ S results in a rather long inertial time period of $T_{in} = 2\pi/f = 2.1$ days. Thus, the sudden changes observed from hydrographic sections B and C within a time span of 6 hours suggest that processes influenced by Earth's rotation, are potentially less important in the final weakening of the temperature front.

Propagating buoyant plumes related to river discharge (Nash and Moum, 2005) or frontal zones in upwelling regimes (Walter et al., 2016) are common dynamical processes in continental shelf regions. In fact, Dale et al. (2008) also identified such a pressure-driven gravity current propagating across the front, once the wind forcing had changed. For our case, using the limited amount of hydrographic information, a propagation speed of $0.4$ m s$^{-1}$ can be identified. This proves to match well with the theorectical estimate for a 2-layer system. The present stratification however complicates the distinction of a sharp

density gradient to apply the gravity current theory. Further, stratified ambient water decreases the propagation speed of a gravity current compared to the two layer system (Ungarish and Huppert, 2002). Still, hinting towards the observation of a gravity current, both transect B and C exhibit strong internal wave signals, which have been observed at the leading edge of such buoyant plumes (Nash and Moum (2005); Bourgault et al. (2016)). Of all mechanisms presented, the gravity current is the only one that may account for the fast changes up to 50 m depth. In contrast, the stability analysis suggests, that MLI are




mainly active in the mixed layer and could only effect lower layers by interacting with the deep mode (Ramachandran et al., 2014; Capet et al., 2016).

In this study we set focus solely on the role of physical processes driving the small-scale distribution of $f\mathrm{CO}_2$. We argue that the timescales considered here are too short to allow for significant contributions of biological processes in driving $f\mathrm{CO}_2$

changes across the front. The strong correlation of surface temperature and $f\mathrm{CO}_2$ imply that the $f\mathrm{CO}_2$ variability is dominated by two processes: Firstly, the temperature dependent solubility of gases in seawater and secondly, warm offshore surface water pushing on top of $\mathrm{CO}_2$-enriched upwelled water, thus creating a mechanical barrier for air-sea gas exchange. However, the weakening of the correlation over time (Fig. 3) indicates that other processes, such as biological activity, might become increasingly important. A more thorough analysis from a biogeochemical perspective is needed, incorporating these effects

(Mahadevan et al., 2004). Consistently, physical processes seem to be able to account for the small scale variability observed by Friederich et al. (2008) and could be crucial to establish a reliable $\mathrm{CO}_2$ budget for the Peruvian upwelling region. Likewise, it is mandatory to have accurate observations of the near-shore wind field, as it proved to be an important factor that contributes to the small-scale evolution of the upwelling front. Modern satellite wind products are still too coarse to resolve the submesoscale frontal variability or other small scale variations such as land-sea breezes or an enhanced near-shore wind stress curl and are

thus not fully reliable within 25 to 50 km from shore (Croquette et al., 2007; Albert et al., 2010).

## 7   Conclusions

High-resolution, underway measurements are demonstrated to be a useful tool in observing the submesoscale variability on scales of $\mathcal{O}(1)$ km. Pronounced changes in the $f\mathrm{CO}_2$ and temperature fields were observed across an upwelling front within hours, providing evidence of high short-term variability in the sea-air $\mathrm{CO}_2$ exchange off Peru. We provide evidence of the

complex submesoscale distribution of surface $\mathrm{CO}_2$ in the Peruvian upwelling sytem and its tight connection to the strong variability in surface temperature. It thus appears that on these timescales the evolving $f\mathrm{CO}_2$ distribution is controlled by physical processes.

Outgassing of $\mathrm{CO}_2$ dropped from $80\,\mathrm{mmol\,m^{-2}\,d^{-1}}$ to less than $10\,\mathrm{mmol\,m^{-2}\,d^{-1}}$ within less than 24 hours. We showed that this drastic change can be explained by physical processes associated with a weakening of the cross-frontal temperature

gradient following a significant decrease in downfrontal windspeed. The initially geostrophically balanced front with a length scale of 10 km vanished within few hours, thereby removing a surface temperature difference of $1°\mathrm{C}$ over 10 km. Hydrographic data shows pronounced changes in the temperature field at depths of up to 50 m. Despite the lack of direct onshore transport of warm water by Ekman dynamics, the wind played a major role in maintaining the front in the beginning of the experiment. The decay of the down-frontal wind gave rise to the development of submesoscale mixed layer instabilities and potentially allowed

for a gravity current to propagate down the cross-frontal pressure gradient. The mixed layer instabilities, however, appear to be too shallow and too slow to be able to account for the complete removal of the cross-frontal temperature gradient. Yet, the onset of a surface gravity current would be consistent with the observed changes in the subinertial period. In addition, our




analysis shows that multiple processes act simultaneously and are likely to interact, thus complicating the identification of a single, dominant mechanism responsible for fast the observed changes in surface $f\mathrm{CO}_2$.

## 8  Data availability

The used data sets are stored in the Kiel Ocean Science Information System (OSIS, https://portal.geomar.de/kdmi, datamanagement@geomar.de) and can be accessed upon request. According to the SFB 754 data policy (https://www.sfb754.de/de/data), all data associated with this publication will be published at a world data center (www.pangaea.de, search projects:sfb754) when the paper is accepted and published.

*Author contributions.*  E. E. Köhn performed the analysis and wrote the manuscript. S. Thomsen designed the experiment and contributed to the manuscript. D. L. Arévalo-Martínez measured and provided the $CO_2$ data and contributed to the manuscript. T. Kanzow was chief scientist on M93, stimulated the analysis and gave scientific guidance. All authors reviewed and commented on the manuscript.

*Competing interests.*  The authors declare that they have no conflict of interest.

*Acknowledgements.*  This work is a contribution of the Sonderforschungsbereich 754 "Climate-Biogeochemistry Interactions in the Tropical Ocean" (www.sfb754.de), which is funded by the Deutsche Forschungsgemeinschaft (DFG). We are grateful to the Peruvian authorities for the permission to carry out scientific work in their national waters. Special thanks go to the captain and the crew of the R/V *Meteor* for their support during the M93 cruise. We are grateful to T. Steinhoff for making available the $CO_2$ sensor as well for its technical support during the cruise. We further thank Liam Brannigan and Leah Johnson for helpful discussions of the results. D. L. Arévalo-Martínez received additional funding support through the BMBF funded SOPRAN II (FKZ 03F0611A) project, as well as the Future Ocean Excellence Cluster at Kiel University (CP0910), and the EU FP7 project InGOS (grant agreement 284274).



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
