# Peer review of "Submesoscale CO2 variability across an upwelling front off Peru"

_Ocean Science, 2017_

## Referee Comment (RC1) · Anonymous Referee #1 · 28 Jul 2017

Kohn et al. observe and address the drivers of CO2 flux variability at small time (hours) and space (km) scales off coastal Peru. Eastern boundary upwelling zones such as this region can be a large source of CO2 to the atmosphere, so understanding the mechanisms that drive CO2 flux from these regions is critical to reducing the uncertainty in the global carbon cycle.

This paper provides an important contribution to understanding the driving mechanisms of CO2 flux from eastern boundary upwelling zones, and I recommend publication in Ocean Sciences after the authors consider some comments below on methodology clarity and how these results relate to other modeling and observational work.

Major comments:

[Figure]

1) Comparison to studies at broader time and space scales: Most observational- and modeling-based studies of CO2 flux do not constrain the influence of submesoscale processes as done here by Kohn et al. In order to make a stronger connection to existing research and provide valuable insight for future studies, this manuscript would benefit from additional analyses that link small to broader scale processes. For example, Kohn et al. compare their observations to the observations presented by Friederich et al. (2008) for the month of February (in section 6). What is the seasonal context for this comparison, i.e., how does February compare to the range of seasonal patterns in CO2 flux and wind forcing in this region? Are submesoscale processes more dominant during certain times of year or certain phases of ENSO? In addition, there have been more recent assessments of broad-scale CO2 flux (Landschützer et al., 2014) that likely utilize more recent underway observations off coastal Peru (Bakker et al., 2014). How do these compare to Friederich et al. and the results presented here? Finally, are there lessons learned as a result of this research on submesoscale processes that may be useful to improving observational design and model parameterizations?

2) Methodology: Further clarification is needed to better understand the experimental design. The beginning of section 2 Data and methods requires a description of the location and timing of field work. This could be addressed by simply moving existing text from the introduction (page 2 lines 21-22) and the results (page 7 lines 2-12). Transects A-C and how they relate to zonal transects 3-7 and cross-frontal transects 1-17) also needs better explanation. Is it possible to show and label all transects in Figures 2 and/or 5 so the reader can better understand how these two figures relate? Transects B and C are mentioned in Fig 2 caption but are not labeled within the figure.

Minor comments/edits:

Page 1 line 5: Here and at key points throughout it would be useful to mention the direction (N-E-S-W) of downfrontal, along-frontal, etc winds.

Page 2 lines 3-5: Some rephrasing necessary here. Processes *are* difficult to observe? By "their importance" do you mean submesoscale processes or models? Do you mean altering, not alterating?

Page 5 section 2.2: What is the estimated uncertainty in fCO2 measurements and resulting CO2 flux? How does using mean salinity during the field study impact the fCO2 uncertainty?

Figure 4: Be consistent in presenting units (e.g., either kg-1 or /kg). State in the caption what the contour lines represent.

Page 7 line 4: Does Feb 16 10:00-15:00 represent the main experiment or the CTD transect?

Figure 5 caption: In panel (e) clarify that velocities are represented by lines and vorticity by circles.

Page 10 line 6: Spell out local time or define acronym.

Page 10 line 25: Is a comma meant after anomalies?

Page 11 line 2: Present consistent # of significant digits.

Page 12 line 8: The weakening of the front is not caused by Ekman transport as demonstrated by what?

Figure 9 caption: Is the uncertainty range represented by the shaded areas around the lines? Add "(green)" after introducing Rb ratio. Why are the Rb ratio errors omitted? Also, consider defining the Rb ratio before presenting Fig 9.

Figure 10 caption: Explain what 200 and 8.7 km represents in panel (a).

Page 17 line 12: Development misspelled.

Page 18 lines 16-17: Wouldn't the difference between delta pCO2 and fCO2 be insignificant compared to measurement uncertainty?

Page 18 line 22: How much does the parameterization of Friederich et al. overestimate

CO2 flux compared to the parameterization used here?

Page 21 line 2: Rephrase "responsible for fast the observed changes"

References:

Bakker, D.C.E., Pfeil, B., Smith, K., Hankin, S., Olsen, A., Alin, S.R., Cosca, C., Harasawa, S., Kozyr, A., Nojiri, Y., O'Brien, K.M., Schuster, U., Telszewski, M., Tilbrook, B., Wada, C., Akl, J., Barbero, L., Bates, N.R., Boutin, J., Bozec, Y., Cai, W.-J., Castle, R.D., Chavez, F.P., Chen, L., Chierici, M., Currie, K., de Baar, H.J.W., Evans, W., Feely, R.A., Fransson, A., Gao, Z., Hales, B., Hardman-Mountford, N.J., Hoppema, M., Huang, W.-J., Hunt, C.W., Huss, B., Ichikawa, T., Johannessen, T., Jones, E.M., Jones, S.D., Jutterstrom, S., Kitidis, V., Koertzinger, A., Landschuetzer, P., Lauvset, S.K., Lefevre, N., Manke, A.B., Mathis, J.T., Merlivat, L., Metzl, N., Murata, A., Newberger, T., Omar, A.M., Ono, T., Park, G.-H., Paterson, K., Pierrot, D., Rios, A.F., Sabine, C.L., Saito, S., Salisbury, J., Sarma, V.V.S.S., Schlitzer, R., Sieger, R., Skjelvan, I., Steinhoff, T., Sullivan, K.F., Sun, H., Sutton, A.J., Suzuki, T., Sweeney, C., Takahashi, T., Tjiputra, J., Tsurushima, N., van Heuven, S.M.A.C., Vandemark, D., Vlahos, P., Wallace, D.W.R., Wanninkhof, R., Watson, A.J. (2014) An update to the Surface Ocean CO2 Atlas (SOCAT version 2). Earth System Science Data 6, 69-90.

Friederich, G.E., Ledesma, J., Ulloa, O., Chavez, F.P. (2008) Air–sea carbon dioxide fluxes in the coastal southeastern tropical Pacific. Progress In Oceanography 79, 156-166.

Landschützer, P., Gruber, N., Bakker, D.C.E., Schuster, U. (2014) Recent variability of the global ocean carbon sink. Global Biogeochemical Cycles, 2014GB004853.
* * *

---

## Referee Comment (RC2) · Anonymous Referee #2 · 29 Jul 2017

This paper offers valuable insight into the impact of eastern boundary upwelling systems on CO2 fluxes. The data is unique and the analysis potentially very useful to the community. I recommend publication after the authors address the comments below.

There is ambiguity and uncertainty with the wind speed and flux estimates that I believe the authors should address:

The authors state that the wind speed was recorded at 35.3 m (Figure 6) and at 35.5 m (section 2.1). Which is it?

The authors state that they followed Smith (1988) to calculate U10 (section 4.5) and also state that Garratt (1997) is used to standardize the wind speed to 10 m height (section 2.2). If the authors are using two different methods, it should be justified. The

method used will certainly impact the estimated 10 m wind speed.

The authors use U10 to determine the wind stress, the Eckman velocity, and the air-sea $CO_2$ fluxes, therefore accurate estimation of the 10 m wind speed is essential to this paper. However, many studies have shown that the boundary layer profile is impacted by the swell/wind sea conditions and their relationship to the wind direction (e.g., Nilsson et al. 2002). Therefore, there is inherent uncertainty in U10 when not accounting these conditions. With a difference between measured U and U10 of over 20 m, there is certain to be error in the U10 estimate which will feed into the parameterizations previously mentioned. This should be addressed.

There is significant uncertainty in the parameterization of the drag coefficient, especially at low wind speeds, i.e. < 5m/s (e.g., Figure 7 of Pan et al., 2005). During this experiment, wind speeds dropped below 5 m/s on several occasions. Therefore inherent uncertainty in estimates of wind stress (Eckman transport) must exist. This should be addressed.

Finally, I wonder if the authors explored flow distortion on the R/V which may also have impacted measured wind speed and therefore U10. This would feed into parameterized variables already mentioned. While the wind direction was generally constant, the R/V travel direction changed throughout the experiment, this would have impacted flow around the R/V, potentially impacting the measured wind speed. This should be addressed.

---

## Author Comment (AC1) · 28 Sep 2017

Dear Editor, dear referees,

in the supplement to this interactive comment you can find the authors' final response to both reviews. Please also find the manuscript changes highlighted in the document.

On behalf of the authors kind regards,

- Eike Köhn

Please also note the supplement to this comment:
https://www.ocean-sci-discuss.net/os-2017-42/os-2017-42-AC1-supplement.pdf

---

## Author Response (AR1)

**Authors' final response on manuscript "Submesoscale CO$_2$ variability across an upwelling front off Peru"**

Eike E. Köhn[1], Sören Thomsen[1], Damian L. Arévalo-Martínez[1], and Torsten Kanzow[2]

[1]GEOMAR Helmholtz Centre for Ocean Research Kiel, Kiel, Germany
[2]Alfred-Wegener-Institute Helmholtz Centre for Polar and Marine Research, Bremerhaven, Germany

*Correspondence to:* Eike E. Köhn (ekoehn@geomar.de)

Dear Editor, dear referees,

we would like to thank you for the evaluation of the manuscript "Submesoscale CO$_2$ variability across an upweling front off Peru". We fully appreciate the generally positive feedback from both referees and the fruitful comments helped to improve the manuscript. This document consists of three parts:

1. In the first part you can find our response to every comment brought forward. It is structured in such a way, that a referee's comment is repeated (blue/red font) with our reaction regarding every issue following below (black font). The reaction is split into the authors' response (AR) and the manuscript changes (MC).

2. The second part consists of the revised manuscript version. The stated page and line numbers in part 1 refer to this version.

3. In part 3 another version of the manuscript is provided showing all the changes made in the document using *latexdiff*. These also include some minor changes to improve the readability of the manuscript.

In the revised version of the manuscript the figure order has been changed by moving Fig. 6 in front of Fig. 4. Now the wind evolution is shown as Fig. 4, the CTD transect as Fig. 5 and the Hovmoeller plots as Fig. 6.

**Anonymous Referee #1:**

Kohn et al. observe and address the drivers of CO2 flux variability at small time (hours) and space (km) scales off coastal Peru. Eastern boundary upwelling zones such as this region can be a large source of CO2 to the atmosphere, so understanding the mechanisms that drive CO2 flux from these regions is critical to reducing the uncertainty in the global carbon cycle. This paper provides an important contribution to understanding the driving mechanisms of CO2 flux from eastern boundary upwelling zones, and I recommend publication in Ocean Sciences after the authors consider some comments below on methodology clarity and how these results relate to other modeling and observational work.

Major comments:

1) Comparison to studies at broader time and space scales: Most observational- and modeling-based studies of $CO_2$ flux do not constrain the influence of submesoscale processes as done here by Kohn et al. In order to make a stronger connection to existing research and provide valuable insight for future studies, this manuscript would benefit from additional analyses that link small to broader scale processes. For example, Kohn et al. compare their observations to the observations presented by Friederich et al. (2008) for the month of February (in section 6). What is the seasonal context for this comparison, i.e., how does February compare to the range of seasonal patterns in CO2 flux and wind forcing in this region? Are submesoscale processes more dominant during certain times of year or certain phases of ENSO? In addition, there have been more recent assessments of broad-scale CO2 flux (Landschützer et al., 2014) that likely utilize more recent underway observations off coastal Peru (Bakker et al., 2014). How do these compare to Friederich et al. and the results presented here? Finally, are there lessons learned as a result of this research on submesoscale processes that may be useful to improving observational design and model parameterizations?

*AR:*    The measurements from the Meteor 93 (M93) research expedition were taken in February 2013, i.e. during austral summer. This period is generally marked by moderate northward wind stress and correspondingly only moderate coastal upwelling. However, the surface Chl-$a$ concentrations and cross-shelf temperature gradients are at their maximum during this period (Echevin et al., 2008; Vazquez-Cuervo et al., 2013). The Takahashi et al. (2014) climatology shows all year round elevated surface $pCO_2$ values in the area spanning from the equatorial Pacific to the Peruvian upwelling region (Fig. AR1). During January/February they are at their minimum, while peak surface $CO_2$ concentrations are observed from August to October. However, the grid (5° in longitude, 4° in latitude) hardly resolves the near-shore region. The increased surface $CO_2$ concentration and the predominantly oceanic $CO_2$ outgassing signal (Takahashi et al., 2009) implicate that the Peruvian upwelling regime is an important region for the air-sea $CO_2$ flux. Only few measurements document the high surface concentrations in the intense coastal upwelling region.

Using the most up to date version of the SOCAT data set (version 3-5) (Bakker et al., 2016) reveals a scarce $f$CO2 data coverage in the Peruvian upwelling region (Fig. AR2). In fact no data is available for the exact region investigated in our

study. Further, Fig. AR3 shows a general lack of data in certain months, including February, the month corresponding to the period of our study. Based on the SOCAT data, it is still not possible to identify a robust mean surface $CO_2$ signal, let alone a seasonal pattern in the coastal Peruvian region. The data set presented in Friederich et al. (2008) is unique in its coverage as well as its temporal and spatial resolution. Most of the comparisons are therefore made with respect to their data set. The

5    newly presented climatology of Landschützer et al. (2017) with a 1° resolution (Fig. AR4) is based on the SOCAT data set and yields a comparable $pCO_2$ pattern as Takahashi et al. (2014). It becomes obvious, that the study area is located at the transition between the high equatorial values and the lower offequatorial concentrations. In Landschützer et al. (2017), the grid point closest to our study region shows a semi-annual cycle in the surface $pCO_2$ content, with its maximum of 440 $\mu$atm in April and September (Fig. AR5). In February $pCO_2$ values are close to the annual average value of about 420 $\mu$atm. However, the

10    scarce data availability in the Peruvian upwelling region shown in Fig. AR3 points towards the large uncertainties associated with the climatology constructed by Landschützer et al. (2017). Further complicating any comparison with the climatology, the conditions observed in our study are not necessarily representative for February. Multiple processes drive the variability in the surface temperature and $f$CO$_2$ field in the Peruvian upwelling region in a broad range of timescales. In February 2013 the Pacific Ocean was goverened by neutral to weak La Niña conditions, associated with a moderately strong upwelling front.

15    These conditions are found to enhance $CO_2$ outgassing in the eastern equatorial Pacific (Feely et al., 2006; Landschützer et al., 2014), and to affect air-sea fluxes in other ocean basins as well (Lefèvre et al. 2013). The impact of the El Niño Southern

[Figure]

**Figure AR1.** Monthly $pCO_2$ climatology of Takahashi et al. (2014) on a 5°×4° grid in $\mu$atm. The red cross shows the location of our study area.

Oscillation (ENSO) on the near-shore $CO_2$ fluxes in the Peruvian upwelling region is still somewhat obscured. Further, the propagation of coastally trapped waves (Pietri et al., 2014) or the curl and variability of the near-coastal wind field (Albert et al., 2010) can largely impact the situation off the coast of Peru. As shown in our study, the location and movement of intensified upwelling fronts is strongly linked to the synoptic wind field. Thus, especially on the observed short timescales, the

5 observations might represent an extreme situation.

The lesson that can be learned from our study is that high variability of surface $CO_2$ induced by physical processes on short timescales can occur in the Peruvian upwelling region. The question whether this variability affects the overall $CO_2$ budget is however not answered. Global data products, as for instance the SOCAT data set, do not have a suffcient coverage of $CO_2$ measurements in this region to allow for analyses that link the local to broader scales.

10 Here is also worth saying that in order to define such fine-scale features such as those described in this study as well as their overall relevance, a way forward is the establishment of multi-platform observation networks in which continuous in situ data is complemented by satellite observations and measurements from autonomous platforms such as e.g. gliders. A successful example of this approach is given by Ohman et al. (2013) who show how linking different platforms helps relating changes in the physical and biogeochemical conditions of the California Current Ecosystem over scales ranging from days to years

[Figure]

**Figure AR2.** The SOCAT surface $fCO_2$ data set shows only scarce data coverage in the Peruvian upwelling region. The number of datapoints collected over 47 years mapped onto a 0.25 degree grid (only for the coastal area) is shown in (a). A 47 year average surface $fCO_2$ distribution on a 1 degree grid constructed from the data in (a) and from data further offshore is given in (b) in $\mu$atm. The red X represents the location of our study.

(and longer). Having such an observational framework is most likely the best way to understand the large impact of short-term variability of an upwelling ecosystem such as the one off Peru. A further example are the moored $CO_2$ observations from Lefèvre et al. (2008) and Lefèvre and Merlivat (2012) which beyond helping to constrain regional budgets and variability of $CO_2$ for the Eastern Tropical Atlantic could also be used to reliably estimate the net carbon community production in this area.

5     We aim to bring attention to the occurrence of submesoscale variability of surface $CO_2$, and how it has the potential to influence our current view of its distribution and emissions from coastal upwelling ecosystems. Upon publication we will ensure the upload of the $CO_2$ dataset to SOCAT.

[Figure]

**Figure AR3.** Monthly composites from 47 years of the SOCATv3 $f$CO$_2$ gridded values (1°) in $\mu$atm.

*MC:*

1. *In Section 6 the following sentence is included (p. 15 l. 15-20): "The conditions observed in this study are not necessarily representative for February conditions. Many processes on different timescales can alter the upwelling frontal structure and intensity off Peru, for instance the state of the El Niño Southern Oscillation (Espinoza-Morriberón et al., 2017) or coastally trapped waves propagating along the Peruvian coast (Pietri et al., 2014). The aim of this study is rather*

[Figure]

**Figure AR4.** Monthly $pCO_2$ climatology of Landschützer et al. (2017) on a 1° grid in $\mu$atm. The magenta cross shows the location of our study area.

[Figure]

**Figure AR5.** Monthly $pCO_2$ climatology of Landschützer et al. (2017) at 13.5°S and 78°5W, i.e. closest grid point to study location.

to analyze a suite of processes involved in the evolution of the upwelling front and surface $f\mathrm{CO}_2$ on short space and timescales ($\mathcal{O}(1$ km$)$, $\mathcal{O}(0.5$ day$)$). "

2. *In Section 7 the following is now included (p. 17 l. 26-31):* "At present, a low $\mathrm{CO}_2$ data coverage within the Peruvian upwelling region (e.g. in SOCAT, Bakker et al. (2016)) complicates the establishment of a reliable climatology, as done by Takahashi et al. (2014) or more recently Landschützer et al. (2017). From our study the importance of the wind and temperature variability on timescales of $\mathcal{O}$(hours) in setting the strength of sea-air $\mathrm{CO}_2$ fluxes becomes obvious. When coupled with $\mathrm{CO}_2$ measurements the use of sea surface temperature and wind products, which capture the high temporal and spatial variability, could lead to improved future estimates of a $\mathrm{CO}_2$ flux climatology off Peru."

3. *The end of Section 7 now reads as (p. 17 l. 31 - p. 18 l. 3):* "In order to understand the large scale impact of short-term variability of an upwelling ecosystem, a way forward would be the establishment of multi-platform observation networks in which continuous in situ data is complemented by satellite observations and measurements from autonomous platforms such as e.g. gliders, as for example conducted by Ohman et al. (2013) in the California Current System. A further example are the moored $\mathrm{CO}_2$ observations from Lefèvre et al. (2008) and Lefèvre and Merlivat (2012) which help to constrain regional budgets and variability of $\mathrm{CO}_2$ for the Eastern Tropical Atlantic and could also be used to reliably estimate the net carbon community production in this area."

2) Methodology: Further clarification is needed to better understand the experimental design. The beginning of section 2 Data and methods requires a description of the location and timing of field work. This could be addressed by simply moving existing text from the introduction (page 2 lines 21-22) and the results (page 7 lines 2-12). Transects A-C and how they relate to zonal transects 3-7 and cross-frontal transects 1-17) also needs better explanation. Is it possible to show and label all transects in Figures 2 and/or 5 so the reader can better understand how these two figures relate? Transects B and C are mentioned in Fig 2 caption but are not labeled within the figure.

*AR:* The Data and Methods part (Section 2) now contains a description of the timing, location and timeline of the experiment. To clarify the experiments procedure, the text was restructured and Figure 2 carries more information regarding the ships route during the cross-frontal transects.

*MC:*

1. *Changes are made to Figure 2 (labeling the individual transect), clarifying the experiment's procedure. The caption now reads as:* "Sampling map of underway temperature (a) and $f\mathrm{CO}_2$ (b) along the cruise track. The solid light grey line shows the CTD section (transect A) conducted prior to the seven zonal transects, with black squares indicating CTD stations. The 17 cross-frontal transects are marked by the dark grey lines (a) and dots (b). The black numbers in (a) correspond to the cross-frontal transects. Transects 12 and 14 correspond to the high-resolution subsurface temperature transects B and C. Magenta arrows show the direction of the ship's track and magenta numbers label the zonal transects. The area presented in both panels is depicted as a red square in Fig. 1."

2. *At the beginning of Section 2.1 the following is included (combining previous text parts from Section 1 and 3) (p. 2 l. 32 - p.3 l. 12):* "Near 14° S and 76°30' W the Peruvian upwelling region is characterized by a strong (quasi-) permanent upwelling cell (Fig. 1). The corresponding "upwelling front" was intensively sampled from February 16 to 18 during the RV *Meteor* cruise M93 in February/March 2013. The field work was carried out within the framework of the Collaborative Research Centre SFB754 "Climate-Biogeochemistry Interactions in the Tropical Ocean" (www.sfb754.de). The experiment's procedure was as follows: Prior to the main experiment, a CTD section (transect "A", black squares in Fig. 2) was conducted on February 16 from 10:00 to 15:00 (time always in UTC unless stated differently) to document the horizontal circulation and the initial vertical distribution of temperature, salinity, dissolved oxygen ($O_2$) and chlorophyll-a (Chl-*a*) across the front. Starting on February 17 at 04:00, the upwelling front was mapped by seven ∼10 km long zonal transects. These were conducted from north to south in 1.8 km spacing (Fig. 2). Each zonal transect took about 45 minutes. From the highest and lowest surface temperature recorded, a cross-frontal axis was estimated. Subsequently, 17 cross-frontal transects were conducted along this axis to study the variability of the front on timescales of several hours. Among these, two high-resolution temperature sections "B" and "C" were conducted as cross-frontal transects 12 and 14. The high-resolution transects took 4.5 hours each to complete, while a regular cross-frontal transect was completed in 40 minutes. The cross-frontal transects were conducted within a 1.8 km wide corridor on three parallel tracks (Fig. 2)."

3. *The end of Section 1 now reads as (p. 2 l. 22-29):* "This paper is structured as follows: In section 2 we present the experiment, the physical and biogeochemical datasets used for this study, as well as the methods employed for their analysis. Section 3 contains a description of the initial state of the front. Observations from the following frontal evolution are presented in section 4. Subsequently, the changes across the temperature front are analyzed in the context of various possible underlying dynamics, such as surface heating (Section 4.1), Ekman buoyancy fluxes (Section 4.2), submesoscale mixed layer instabilities (Section 4.3) or pressure driven gravity currents (Section 4.4). In section 5 these mechanisms are compared with respect to their associated buoyancy fluxes. Section 6 contains a discussion of the different mechanisms which possibly drive the observed variability. The conclusions drawn from this study follow in section 7."

4. *In Section 6 the following is included (p. 16 l. 1-3):* "Further, the fact that the cross-frontal transects show coherent signals, even though not all cross-frontal transects were performed on the exact same track, but about 1 km apart, points towards a weak along-frontal flow variability."

Minor comments/edits:

Page 1 line 5: Here and at key points throughout it would be useful to mention the direction (N-E-S-W) of downfrontal, along-frontal, etc winds.

*AR:* It is dynamically more important that the wind is blowing constantly downfrontal than its absolute direction, but at key points throughout the manuscript the direction (e.g. "equatorward") is added to clarify the meaning of along-frontal or

downfrontal.

Page 2 lines 3-5: Some rephrasing necessary here. Processes \*are\* difficult to observe? By "their importance" do you mean submesoscale processes or models? Do you mean altering, not alterating?

*AR:* This part is rephrased for more clarity. The typo regarding "altering" is also corrected.

*MC: In Section 1 the rephrased sentence reads as (p. 2 l. 2-5):* "However, submesoscale frontal processes are difficult to observe due to their small spatial and temporal scales. At the same time the importance of these dynamics for physical-

10  biogeochemical coupling has been put forward by model studies, e.g. by altering the vertical transports of nutrients and organic carbon ..."

Page 5 section 2.2: What is the estimated uncertainty in fCO2 measurements and resulting CO2 flux? How does using mean salinity during the field study impact the fCO2 uncertainty?

*AR:* The uncertainty of the $CO_2$ measurements was about +/- 2 $\mu$atm. The $f CO_2$ is mostly dependent on temperature with an isochemical dependence of 15 $\mu$atm per °C (this means that a 1 °C error in temperature measurements can lead to a bias as high as 15 $\mu$atm in $f CO_2$, see e.g. Körtzinger et al. (2000); Pierrot et al. (2009)). The effect of salinity is rather marginal in comparison with the one of temperature both for the $f CO_2$ computation as well as for the flux estimates since both viscosity and

20  the molecular diffusion coefficients used for the calculations vary only slightly with salinity changes (see eg. Pilson (2012)). To illustrate this point we calculated the $f CO_2$ with $s_{34}$ = 34.0406 and $s_{36}$ = 36.0406 and compared the mean differences to the values we used for this study. As can be seen in Figure AR6, changing salinities did not considerably affect $f CO_2$ values (mean difference for $s_{34}$ = 0.0055 and for $s_{36}$ = -0.0055).

25  *MC: In Section 2.2 the following sentence is added (p. 6 l. 11-14):* "A change in the mean salinity by 1 leads to a mean off-set of 0.0055 $\mu$atm and has thus a small influence compared to temperature with an isochemical dependence of 15 $\mu$atm per °C (e.g. Körtzinger et al. (2000); Pierrot et al. (2009)). The uncertainty of the $CO_2$ measurements was hence about +/- 2 $\mu$atm."

Figure 4: Be consistent in presenting units (e.g., either kg-1 or /kg). State in the caption what the contour lines represent.

*AR:* The units of Fig. 5 (previously Fig. 4) are changed to consistently use the exponential form. It is stated in the caption, that the white contour lines represent isopcynals calculated from the CTD data. The thin white lines present before due to rendering issues are removed in the revised version.

[Figure]

**Figure AR6.** Influence of salinity on $f\mathrm{CO}_2$ during the entire M93 expedition.

Page 7 line 4: Does Feb 16 10:00-15:00 represent the main experiment or the CTD transect?

*AR:* The time information given in parantheses is shifted further back in the sentence to clarify the association of this time period with the CTD transect.

*MC: The sentence is now included in Section 2.1 and reads as follows (p. 3 l. 3-5):* "Prior to the main experiment, a CTD section (transect "A", black squares in Fig. 2) was conducted on February 16 from 10:00 to 15:00 (time always in UTC unless stated differently) to document the horizontal circulation and the initial vertical distribution of temperature, salinity, dissolved oxygen ($O_2$) and chlorophyll-a (Chl-*a*) across the front."

Figure 5 caption: In panel (e) clarify that velocities are represented by lines and vorticity by circles.

*AR:* Symbols are clarified.

15     *MC: The beginning of the caption of Fig. 6 (previously Fig. 5) now reads as:* "Hovmoeller diagrams of the cross-frontal surface temperature gradient (a), underway surface temperatures (b), surface $f\mathrm{CO}_2$ (c), ocean-atmosphere $CO_2$ fluxes (d), mean current velocities in the upper 40 m (black lines) and vorticity (colored circles) (e) and diabatic surface heating (f) for the last

five zonal transects and the subsequent 17 cross-frontal transects."

Page 10 line 6: Spell out local time or define acronym.

*AR:* LT changed to "local time".

*MC:*

1. *The sentence in Section 4.1 now reads as (p. 9 l. 9-10):* "The wind begins to drop at 11:00 on February 17 (06:00 local time) and the temperature gradient fully vanishes 14 hours later at 01:00 on February 18 (Fig. 6a)."

2. *In the description of the diurnal cycle of surface heat fluxes (Section 4.1) the local time is added at a few instances for better readability.*

*AR:* comma inserted.

*MC: The sentence in Section 4.1 now reads as (p. 9 l. 29-30):* "Even though this roughly corresponds to the amplitude of the temperature anomalies, the lateral homogeneity of the net surface heat flux prohibits to attribute the development of the anomalies to surface heating."

Page 11 line 2: Present consistent # of significant digits.

*AR:* The salinity is changed to be consistently given with one digit.

*MC: The sentence in Section 4.1 now reads as (p. 10 l. 5-6):* "The inital CTD section shows a surface salinity varying between 35.1 and 35.2 (Fig. 5c)"

Page 12 line 8: The weakening of the front is not caused by Ekman transport as demonstrated by what?

*AR:* This sentence is restructured to clarify its function of summarizing the above results.

*MC: The sentence in Section 4.2 now reads as (p. 11 l. 11):* "The results above show, that the weakening of the temperature front is not directly caused by onshore Ekman transport."

Figure 9 caption: Is the uncertainty range represented by the shaded areas around the lines? Add "(green)" after introducing Rb ratio. Why are the Rb ratio errors omitted? Also, consider defining the Rb ratio before presenting Fig 9.

*AR:* It is now clarified in the caption that the uncertainty range is represented by the shaded areas. The error is also included
5  for the Rb ratio. Introducing the Rb ratio before referencing the Fig. 9 would be helpful, but it is only important in the comparison of buoyancy fluxes and could lead to confusion if introduced while discussing the individual physical processes.

*MC: The caption of Figure 9 now reads as:* "Vertical buoyancy fluxes associated with surface heating (blue), Ekman transport (red) and mixed layer instabilities (magenta). The shaded areas around the Ekman buoyancy flux (EBF) and the buoyancy
10  flux associated with mixed layer instabilities (MLI) are accounting for uncertainties in the surface salinity gradient (see text for more details). The shaded uncertainty range for vertical buoyancy fluxes due to surface heating is given as one standard deviation obtained from sectional averaging. The sum of the three processes is given by the black line. The uncertainty is given as the maximum error resulting from the three processes. The $R_b$ ratio (green) and its uncertainty are treated similarly. For cross-frontal transect 13 and 14 no $R_b$ value or error can be stated due to the vanishing denominator in the definition of $R_b$.
15  $\log_{10}(R_b) > 0$ points towards a net restratification, while $\log_{10}(R_b) < 0$ indicates a destabilization of the water column. For calculation details of all quantities see text. Time is given in UTC. The small black numbers on top indicate the respective zonal or cross-frontal transect number."

Figure 10 caption: Explain what 200 and 8.7 km represents in panel (a).

*AR:* It is now highlighted in the caption of Fig. 10, that the labels show the length scales associated with the local maxima for better readability. The length scales are now also explicitly stated in the figure caption in brackets.

*MC: The caption of Figure 10 now reads as:* "(a) Initial growth rate of barolinic instabilities obtained from linear stability
25  analysis, applied to the frontal state during the CTD section (transect A). The local maxima of the growth rate are labeled by their corresponding length scale in $\mathrm{km}$. The smoothed stratification and geostrophic along-frontal velcity profiles used as the background state, are shown in panels (b) and (c). A deep mode at large horizontal scales (200 $\mathrm{km}$) and a shallow mode, associated with mixed layer instabilities and short horizontal length scales (8.7 $\mathrm{km}$) are present. The corresponding anomalies for the cross-front (u) and along-front (v) velocities are shown in panels (d) and (e) for the deep and shallow modes, respectively."

*AR:* Typo corrected.

*MC: The sentence in Section 5 now reads as (p. 13 l. 31 - p. 14 l. 1):* "During the time of the development of the coherent surface temperature anomalies (cross-frontal transects 7-11), the combination of the three processes favor a restratification of the mixed layer (Fig. 9)."

 Wouldn't the difference between delta pCO2 and fCO2 be insignificant compared to measurement uncertainty?

*AR:* Normally the difference between $p\mathrm{CO_2}$ and $f\mathrm{CO_2}$ is about 3%. However regardless of the uncertainty of Friederich et al. (2008) estimates, we opted for using the same magnitude in order to compare both studies.

*MC: The sentence in Section 6 now reads as (p. 15 l. 4-6):* "Our observations show $\mathrm{CO_2}$ outgassing rates between 3.5 and 30 $\mathrm{mol\ m^{-2}\ yr^{-1}}$ which are in line with the outgassing signals observed by Friederich et al. (2008), even after accounting for the difference in converting our values from partial pressures to fugacities (difference usually about 3%)."

 How much does the parameterization of Friederich et al. overestimate CO2 flux compared to the parameterization used here?

*AR:* The Wanninkhof (1992) (W92) parameterization which was used on the study by Friederich et al. (2008) is known to overestimate the gas transfer velocities (and therefore the air-sea fluxes; see Garbe et al. (2014)). Since the authors of that study used a slightly different methodology to ours and employed monthly wind products instead of instantaneous, it is difficult to provide an accurate estimate of how much their values overestimate $\mathrm{CO_2}$ with respect to ours, which are based on the Nightingale et al. (2000) parameteriaztion (N00). However, computing our fluxes with both parameterizations resulted in a mean flux density difference (W92-N00) of 0.802 $\mathrm{mmol\ m^{-2}\ d^{-1}}$, which in turn equates to about 17% difference. Hence, had we used the same parameterization as Friederich et al. (2008), our flux density calculations would have resulted in even higher values compared to those from their study.

*MC: The part in Section 6 now reads as (p. 15 l. 9-13):* "The results from Friederich et al. (2008) were re-scaled to a much larger area (5° S - 15° S), have been subject of spatial extrapolation and smoothing, and were calculated with the sea-air gas exchange parameterization by Wanninkhof (1992) which tends to overestimate the gas transfer velocities (see Wanninkhof (2014)). Simply using the Wanninkhof (1992) parameterization instead of the Nightingale et al. (2000) parameterization yields 17% higher fluxes, already indicating that it is difficult to draw a direct comparison between both data sets."

 Rephrase "responsible for fast the observed changes"

*AR:* Swapped the words "fast" and "the" to correct the sentence.

*MC: The sentence in Section 7 now reads as (p. 17 l. 24-25):* "In addition, our analysis shows that multiple processes act simultaneously and are likely to interact, thus complicating the identification of a single, dominant mechanism responsible for the fast observed changes in surface $f\mathrm{CO_2}$."

wind coming from within a corridor of +/- 60° around perfect bow-on wind. Fig. AR8b shows that $\sim 35\%$ of the wind data were measured in a wind corridor of +/- 60°, but that $\sim 80\%$ of the wind data was measured in a somewhat wider wind corridor of +/- 90°.

While the ship's flow distortion produces some error in the estimation of the wind speed through deceleration/acceleration and vertical displacement, it is unlikely that it can obscure the dominant signal of the wind speed decrease. The high and exposed location of the measurement platform are expected to keep the error at a minimum level.

*MC: No manuscript changes.*

[Figure]

**Figure AR8.** a) Timeseries of the ship's heading (black) during the zonal and cross-frontal transects. 270° correspond to westward heading, 180° to southward and 90° to eastward heading. The direction of the predominantely southerly (180°) winds is given in blue, which is approximately parallel to the along-frontal axis (red). Throughout the transects there exists a difference between the wind direction and heading of around 90°, indicating their orthogonality. b) Histogram of the relative wind direction. A direction of 0°/360° correponds to head wind, while 180° corresponds to tail wind and 90° corresponds to wind coming from starboard.

[revised manuscript text omitted]